# Dynamics of collective cooperation under personalised strategy updates

Yao Meng [1], Sean P. Cornelius[2], Yang-Yu Liu [3,4] & Aming Li [1,5] ✉

Collective cooperation is essential for many social and biological systems, yet understanding how it evolves remains a challenge. Previous investigations report that the ubiquitous heterogeneous individual connections hinder cooperation by assuming individuals update strategies at identical rates. Here we develop a general framework by allowing individuals to update strategies at personalised rates, and provide the precise mathematical condition under which universal cooperation is favoured. Combining analytical and numerical calculations on synthetic and empirical networks, we find that when individuals' update rates vary inversely with their number of connections, heterogeneous connections actually outperform homogeneous ones in promoting cooperation. This surprising property undercuts the conventional wisdom that heterogeneous structure is generally antagonistic to cooperation and, further helps develop an efficient algorithm OptUpRat to optimise collective cooperation by designing individuals' update rates in any population structure. Our findings provide a unifying framework to understand the interplay between structural heterogeneity, behavioural rhythms, and cooperation.

Cooperative behaviour—in which individuals pay a cost to confer a benefit to others—is widely and deeply embedded in human and animal societies alike, and has attracted great research interests in studying the underlying mechanisms of favouring the emergence of cooperation[1–15]. Under the prominent metaphor of the prisoner's dilemma[16], without additional mechanisms including direct[17,18] or indirect reciprocity[19–22], and punishment[23,24], unstructured populations —wherein everyone interacts with everyone else—are known to leave no opportunity for the survival of cooperators[25,26]. Thus in recent decades, researchers have been exploring evolutionary game dynamics in structured populations, where who interacts with whom is determined by a network or population structure, with links representing interactions between different individuals (nodes)[4–6,27–30]. The central question is: which population structures promote cooperation, and which hinder it?

In homogeneous networks—where all individuals basically have similar numbers of neighbours—a well-known finding is that cooperation is favoured if the ratio between the benefit ($b$) provided by a cooperator and the associated cost paid ($c$) exceeds the average number of neighbours $\langle k \rangle$, namely the simple rule[4] $b/c > \langle k \rangle$. Similar results can be found in the more general case: Allen et al. analytically calculated the critical benefit-to-cost ratio $C^*$, above which cooperation is promoted for an arbitrary network topology[5]. Apart from confirming $C^* = \langle k \rangle$ for homogeneous structures, this result informs a higher value of $C^*$ for heterogeneous structures[31], wherein different individuals may have wildly different numbers of neighbours. Accordingly, although heterogeneous structures like scale-free networks[32] are ubiquitous in real systems, they appear to hinder the emergence of cooperation compared to homogeneous structures[31].

[1]Center for Systems and Control, College of Engineering, Peking University, Beijing 100871, China. [2]Department of Physics, Toronto Metropolitan University, Toronto, ON M5B 2K3, Canada. [3]Channing Division of Network Medicine, Department of Medicine, Brigham and Women's Hospital and Harvard Medical School, Boston, MA 02115, USA. [4]Center for Artificial Intelligence and Modeling, The Carl R. Woese Institute for Genomic Biology, University of Illinois at Urbana-Champaign, Champaign, IL 61801, USA. [5]Center for Multi-Agent Research, Institute for Artificial Intelligence, Peking University, Beijing 100871, China. ✉e-mail: amingli@pku.edu.cn

Despite remarkable advances in our understanding of the emergence of cooperation, many studies have confined that individuals update their strategy synchronously[6,33–35]—that all individuals update at exactly the same time. However, perfect synchronism is absent from the real world, and it has been shown that asynchronous updating—individuals are allowed to update at different time—can significantly alter the evolution of cooperation[36–40]. A typical asynchronous update rule is the death-birth update, where only a single individual is selected uniformly at random to die and their neighbours spread their strategies by competing for the vacant position at each time step[4,5]. Alternatively, individuals may change their strategies by mimicking that of their neighbours (imitation[4], pairwise comparison[12,41]). All these important canonical updating rules have been based on a key assumption: that all individuals update their strategies at the same rate.

In reality, humans behave in more sophisticated ways in decision-making than simple identical updating. An empirical study on evolutionary games uncovered that individuals are observed to have many different possibilities for strategy updating in human behavioural experiments[42]. Indeed, both cognitive processing speed and personality traits can have an impact on the time of individual decision-making. Previous empirical studies have found that individuals vary significantly in cognitive processing speed[43–45]. For example, individuals with greater cognitive abilities have high information processing speed and display a short reaction time. On the other hand, many personality traits are also evidenced to correlate with the decision-making speed[46]. Taken together, the previous assumption of identical update rates for all individuals is too ideal to portray the update event and heterogeneous individual interaction rhythms in realistic scenarios[47,48]. This prompts us to ask how this dynamical heterogeneity might interact with structural heterogeneity to alter the evolution of cooperation.

Here we investigate evolutionary game dynamics under non-identical rates of strategy updating. Specifically, we consider the scenario where individuals are allowed to update their strategies at diverse individual rates. We find that non-uniform rates of strategy updating can have profound effects on the emergence of cooperation, especially on heterogeneous structures, and reveal a significant decrease in $C^*$ necessary to promote cooperation. Moreover, we develop an efficient algorithm OptUpRat to minimise the threshold for the emergence of cooperation by tuning the update rate of each individual on any network.

## Results

We consider evolutionary game dynamics on a structured population of $N$ players, whose interactions are represented by an undirected, unweighted network. At any given time, the state of each node (player) is characterised by a strategy of either cooperation (C) or defection (D) (Fig. 1a). In each round of the game, every node $i$ plays the game pairwise with its immediate $k_i$ neighbours. Specifically, cooperators pay a cost $c$ to provide a benefit $b$ to each of their neighbours, while defectors pay nothing, and thus provide no benefit. In this way, each node $i$ gains an average payoff $f_i$, corresponding to the average benefits received (from neighbouring cooperators) minus its cost.

Traditionally, individuals are assumed to update their strategies following independent Poisson processes with identical rates. Here we depart from this practice: allowing each individual $i$ to update its strategy with personalised rate $\lambda_i$ (Fig. 1b). When an individual is chosen for an update, it does so by copying the strategy of one of its neighbours $j$, with probability proportional to the fitness of $j$, generally defined as $F_j = 1 + \delta f_j$, where $\delta > 0$ captures the intensity of selection[4,5] (see Methods). For strong selection intensity, cooperation is disfavoured since the initial cooperator will not be able to survive

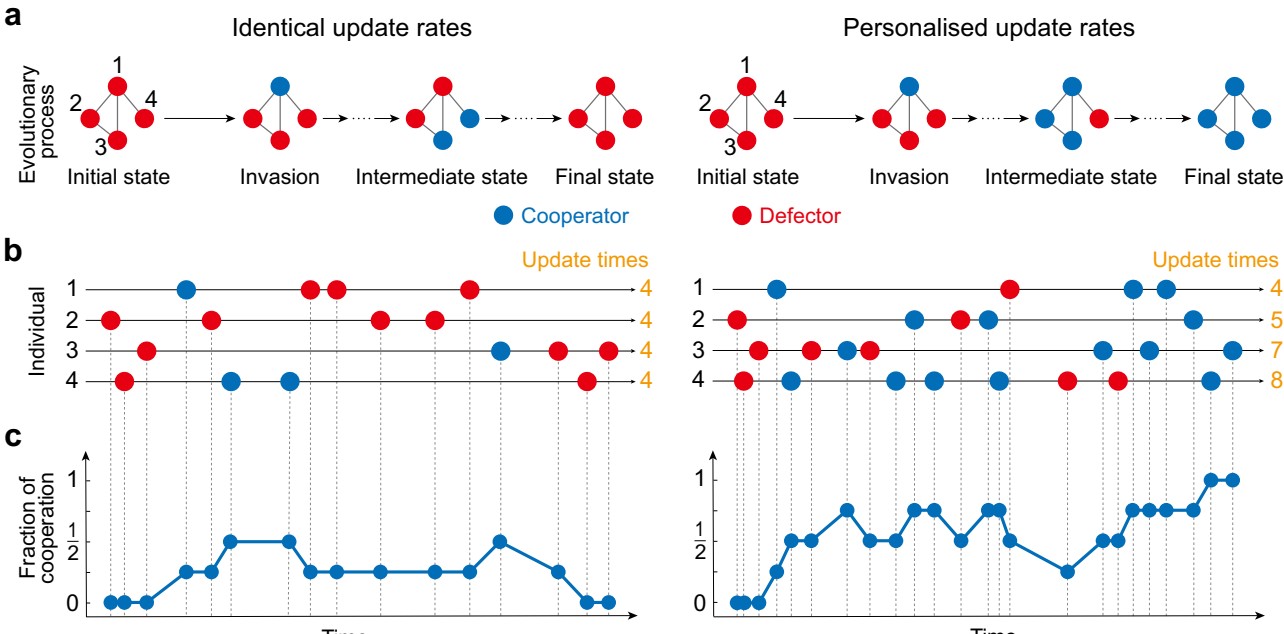

**Fig. 1 | Illustration of the evolutionary process with identical versus personalised rates of strategy update.** The interactions between four individuals are depicted in the example network structure in **a**, where individuals play games with their neighbours and gain the corresponding payoffs. The evolutionary process starts from a population of full defectors (red), and a cooperator (blue) invades the population via the top site. **b** The update event for each individual occurs as a Poisson process. We indicate on the timeline when each individual is chosen to update its strategy. The colour of the dot indicates the strategy after the update,

which may be unchanged. When individuals' update rates are identical, they will have approximately the same number of strategy updates (numbers in orange, left panel), while for non-identical update rates, individuals with higher rates will update their strategies more often (right panel). The update rates for each individual in the right panel are $\lambda_1 = 1$, $\lambda_2 = 1.25$, $\lambda_3 = 1.75$ and $\lambda_4 = 2$, respectively. The change in the fraction of cooperation throughout the game is illustrated in **c**, and the evolutionary process ends when the population reaches a state of either full defection (left panel) or full cooperation (right panel).

or spread its strategy. Thus, to systematically uncover the effects of heterogeneous update rates on the fate of cooperators compared to existing findings, here we focus on the canonical case of weak selection.

To quantify the ability of cooperation to proliferate, we initialise our simulations with a single cooperator placed uniformly at random in a population among $N-1$ defectors. The evolutionary game ends when a state with either all cooperators or all defectors is reached (Fig. 1c). We define the fixation probability of cooperation ($\rho_C$) as the probability of reaching the state of full cooperation over many realisations of this process. We can analogously define a probability $\rho_D$ of reaching a full-defection state starting from a single defector planted of $N-1$ cooperators. Our interest in this study is the condition under which cooperation is favoured to replace defection than vice versa[4,5,26], namely $\rho_C > \rho_D$. This condition is equivalent to $\rho_C > 1/N$ (Supplementary Note 1), namely that selection favours the emergence of cooperation relative to the neutral drift ($\delta = 0$), in which neither cooperation nor defection is favoured ($\rho_C = \rho_D = 1/N$).

## Evolutionary game dynamics on complex networks

First, we explore how the heterogeneous strategy updating affects the fate of cooperators on four commonly-studied population structures: lattice, small-world, Erdös-Rényi, and scale-free networks (Fig. 2). Under the traditional scenario of identical update rates ($\lambda_i = 1$ for all $i$), scale-free networks demand the largest critical benefit-to-cost ratio $C^*$, above which cooperation is favoured among all the four structures, and the lattice structure the smallest (Fig. 2a), consistent with previous findings[4,5]. But surprisingly, when a node's update rate varies inversely with its number of neighbours ($\lambda_i = 1/k_i$), we find that this trend is reversed (Fig. 2b). Here, the scale-free network becomes the most amenable to cooperation, and lattice the least. Interestingly, we find that heterogeneous update rates can even improve upon the canonical threshold $b/c > \langle k \rangle$ (namely, $C^* = \langle k \rangle$) for homogeneous populations[4], allowing cooperation to emerge even when $b/c < \langle k \rangle$ (namely, $C^* < \langle k \rangle$). Furthermore, we find that this pattern is strengthened when the update rate is inversely proportional to higher powers of $k_i$ (Fig. 2d). In contrast, when $\lambda_i$ is positively related to $k_i$, the ordering of $C^*$ over

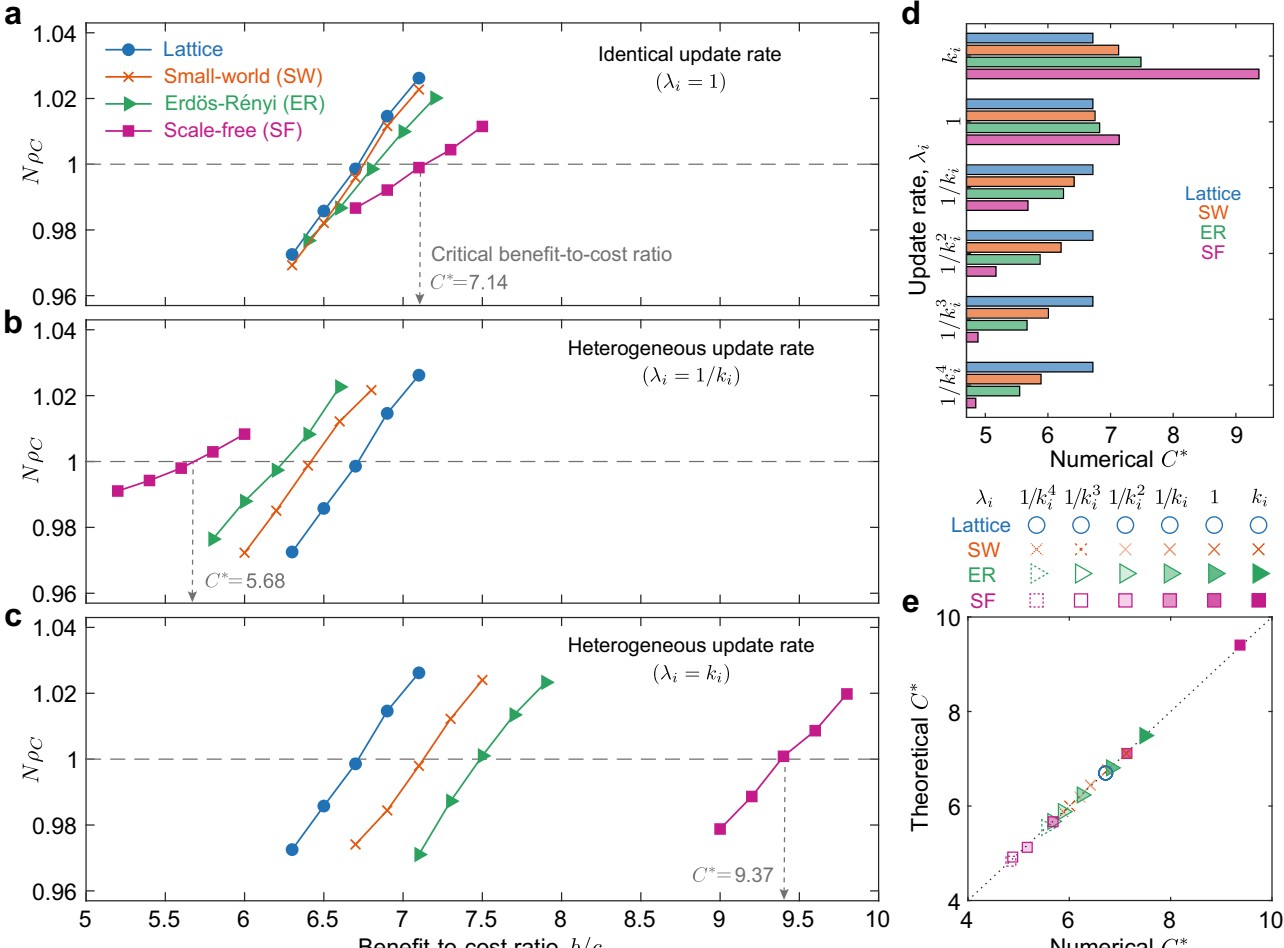

**Fig. 2 | Effect of heterogeneous strategy update rates on the emergence of cooperation.** We show the fixation probability of cooperation ($\rho_C$) as a function of the benefit-to-cost ratio ($b/c$) over different settings of the update rate ($\lambda_i$) of individual $i$, namely identical ($\lambda_i = 1$ for every individual in **a**) and heterogeneous ($\lambda_i = 1/k_i$ in **b** where $k_i$ is the number of neighbours of $i$, $\lambda_i = k_i$ in **c**) on lattice, Erdös-Rényi[65] (ER), small-world[66] (SW) and scale-free[32] (SF) networks, respectively. The critical benefit-to-cost ratio $C^*$ above which the cooperation is favoured for each network occurs when the corresponding curve intersects the horizontal line representing the neutral-drift case ($\rho_C = 1/N$). $C^*$ for the scale-free case (purple) is marked. We demonstrate that the trend of $C^*$ reverses when the update rate varies inversely with $k_i$ in **b**, presenting the advantage of SF networks on favouring cooperation. **d** The ordering of $C^*$ for the four networks considered holds with $\lambda_i = 1/k_i^\gamma$ ($\gamma = 1, 2, 3, 4$). Here we also show that SF networks are the most amenable to cooperation at non-identical update rates compared with other networks. **e** Simulation results on $C^*$ in **a**–**d** are in good agreement with our theoretical calculations shown in equation (1). Numerical values of $\rho_C$ are obtained from the fraction of simulations in which the population reaches full cooperation out of $10^7$ independent realisations on networks of 98 nodes for lattice and 100 for other networks with an average degree $\langle k \rangle = 6$, and $\delta = 0.01$. Source data are provided as a Source Data file.

different structures matches the identical-rate case, but with the inhibition of cooperation fixation by heterogeneous networks amplified (Fig. 2c). We show the robustness of our results over different population sizes, average degrees and selection intensities in Supplementary Figs. 1–3.

We further shed light on our numerical findings by deriving a closed-form expression for the critical benefit-to-cost ratio $C^*$ as a function of the network structure (see Methods)

$$C^* = \frac{\sum_{i,j} k_i p_{ij}^{(2)} \eta_{ij}}{\sum_{i,j} k_i p_{ij}^{(3)} \eta_{ij} - \sum_{i,j} k_i p_{ij} \eta_{ij}}. \tag{1}$$

Here, $k_i = \sum_j e_{ij}$ defines the number of neighbours (degree) of individual $i$, and $e_{ij} = e_{ji} = 1$ indicates that there is an edge between nodes $i$ and $j$ ($e_{ij} = e_{ji} = 0$ otherwise). The probability of a 1-step ($n$-step) random walk from $i$ to $j$ is denoted by $p_{ij}$ ($p_{ij}^{(n)}$), and $\eta_{ij}$ is the coalescence time[49]—the expected time for two random walks starting from nodes $i$ and $j$ to meet at a common node. As shown in Fig. 2e, all numerical results in Fig. 2a-d are in good agreement with the theoretical prediction of equation (1).

## Role of network hubs

To intuitively understand why heterogeneous update rates can improve the fixation of cooperation in heterogeneous networks, we first consider how the evolutionary dynamics play out on a simple double-star structure (Fig. 3). When the fixation of cooperation occurs in this highly heterogeneous structure, it usually does so in four stages: (I) occupation of one of the hubs; (II) formation of a stable cluster of cooperators among that hub and its neighbours; (III) occupation of the other hub; and finally (IV) spread to the remaining nodes. As such, the ultimate triumph of cooperators can be thwarted if a hub imitates defection from even one of its (many) neighbours before stages (II) and (IV) are complete (Fig. 3c). There are ample opportunities for this to occur under the traditional setting of identical update rates ($\lambda_i = 1$), as illustrated in Fig. 3b. When $\lambda_i = 1/k_i$ however (Fig. 3a), hubs update relatively infrequently. As such, once a hub becomes a cooperator, it is effectively locked-in, giving time for its strategy to spread to the hub's

neighbours. Note that this lock-in effect can facilitate the formation of cooperative clusters to have higher payoffs to resist the invasion of defectors, yet defectors receive a lower payoff after driving their neighbours to defectors and further reduce their survival chances. By the same logic, the preferential updating of hubs ($\lambda_i = k_i$) usually leads to the extinction of cooperation, as the formation of stable clusters of cooperators and the spread of cooperation is even harder than the traditional scenario of identical updating (Fig. 3c).

In Fig. 4, we illustrate the fundamental mechanism explaining why infrequent updates of hubs can facilitate cooperation. If an individual (grey node in Fig. 4a) decides to update its strategy, it will imitate the strategy of its neighbours according to their payoffs. The neighbouring cooperator obtains an average payoff $P_C = bq_{C|C}(\langle k \rangle - 1)/\langle k \rangle - c$ and the neighbouring defector obtains $P_D = bq_{C|D}(\langle k \rangle - 1)/\langle k \rangle$, where $q_{C|C}$ ($q_{C|D}$) represents the conditional probability to find a cooperative neighbour for a given cooperator (defector). The contribution to the neighbouring cooperator and defector from the updating individual is excluded since they are equal. Thus the cooperator is favoured compared to the defector to disperse its strategy if $P_C > P_D$, namely

$$b(q_{C|C} - q_{C|D})(\langle k \rangle - 1)/\langle k \rangle - c > 0, \tag{2}$$

with $Q = (q_{C|C} - q_{C|D})(\langle k \rangle - 1)$ capturing the average number of cooperative neighbours that a cooperator has more than a defector. For the canonical setting with identical update rates ($\lambda_i = 1$), we know $Q = 1$ according to pair approximation (Supplementary Note 2), namely a cooperator has on average one more cooperative neighbour than a defector (Fig. 4a). This leads to the conclusion that cooperation is favoured when $b/c > \langle k \rangle$ (namely, $C^* = \langle k \rangle$), which also degenerates to the simple rule[4] for homogeneous networks where $k_i = \langle k \rangle$.

Next we show how heterogeneous update rate alters the local dispersal of cooperation. When $\lambda_i = 1/k_i$, we find that $Q > 1$ (Supplementary Note 2), indicating that the number of cooperative neighbours of a cooperator exceeds that of a defector by more than one (Fig. 4b). This implies that the net payoff of cooperators relative to defectors is further increased, giving cooperators more advantage in competition and dispersal. Therefore, the critical ratio for $\lambda_i = 1/k_i$ is

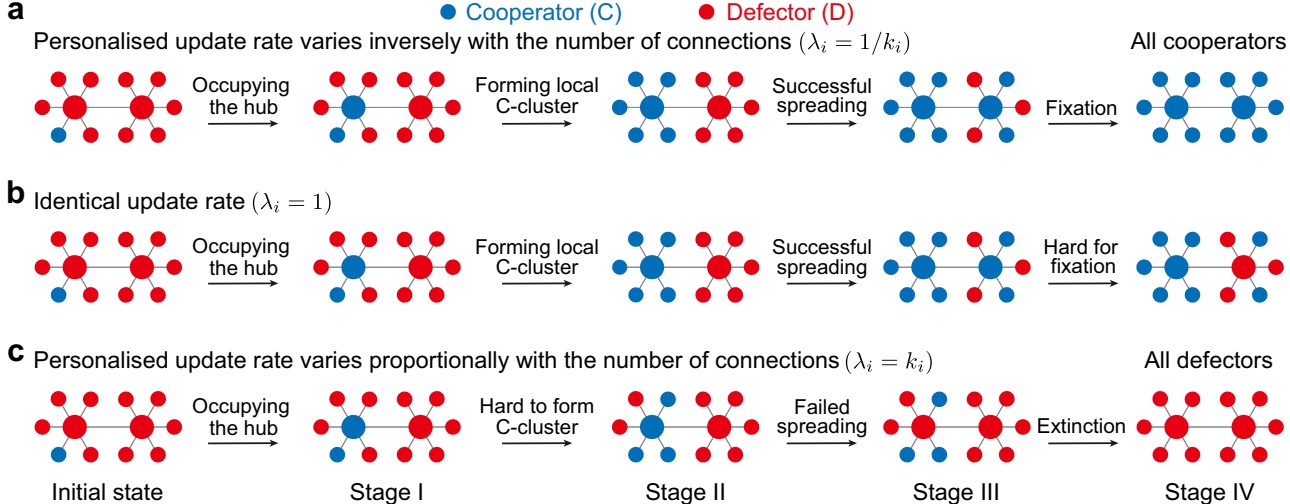

**Fig. 3 | Illustration of the role of hubs in the evolution of cooperation on a double-star structure.** **a** The hubs, two centres of the double-star structure for example, have low update rates when $\lambda_i = 1/k_i$ ($k_i$ is the number of connections for each node), which facilitates the formation of local cluster of cooperation (blue dot, Stage II) once it is occupied by a cooperator (Stage I). Likewise, once the left hub spreads cooperation to the right hub (Stage III), the remaining nodes are quickly

driven to cooperators (Stage IV). **b** When the update rates are identical ($\lambda_i = 1$), the hubs have many opportunities to change their strategy to defection before all neighbours become cooperators (Stage IV), making the fixation of cooperation less likely. **c** The hub switches its strategy quite frequently when $\lambda_i = k_i$, which makes it hard to form even the left C-cluster (Stage II), to say nothing of spreading cooperation to the right centre.

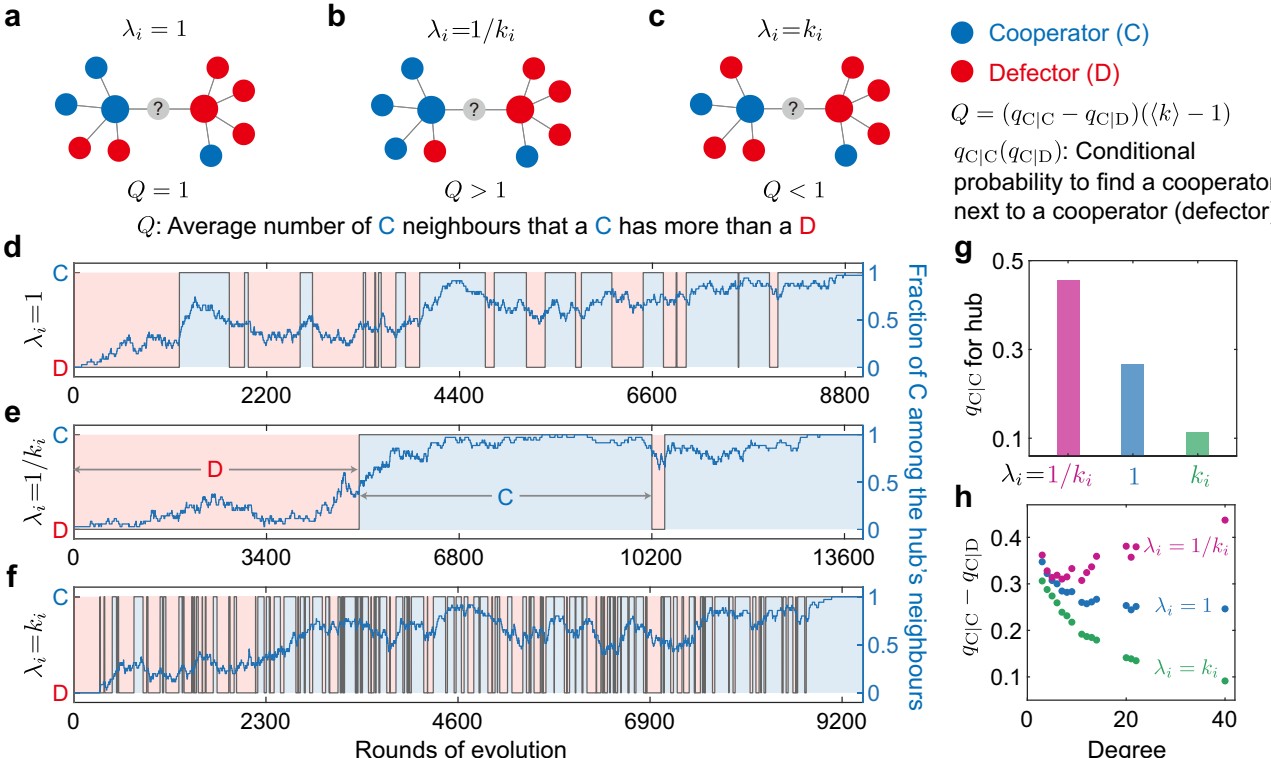

**Fig. 4 | Mechanism for promoting collective cooperation with infrequent strategy updates of hubs. a–c** Illustration on the scenario where a cooperator (blue dot) and a defector (red dot) compete to spread their strategy to the individual (grey dot) selected for strategy update under different update rates $\lambda_i$. Since behaviour dispersal occurs in the neighbourhood, the cooperator obtains on average $bQ/\langle k \rangle - c$ more payoff than the defector (equation (2)), and the cooperator is favoured when the above expression is positive. **a** For identical updating ($\lambda_i = 1$), the cooperator has one more cooperative neighbour than the defector, therefore it receives $b/\langle k \rangle$ more benefit than the defector at a cost of $c$. **b** When $\lambda_i = 1/k_i$, the net benefit of the cooperator relative to the defector exceeds $b/\langle k \rangle$ because the fraction of cooperative neighbours of the cooperator further increases compared to the defector, offering the cooperator a higher chance for dispersal. **c** We show that the fast strategy update of hubs ($\lambda_i = k_i$) reduces the number of cooperative neighbours

of the cooperator, which exceeds that of the defector by less than one. This lowers the benefit of the cooperator and reduces the chance to win the empty site. We further compare the state of the hub (grey lines) and the fraction of cooperation among its neighbours (blue lines) of a scale-free network with different settings of update rates (**d–f**). Generally, the hub imitates one of its cooperative neighbours and keeps cooperation for several rounds (light blue shaded region) before switching to defection (light red shaded region) in **d**. Statistically, we count the fraction of cooperators in the neighbourhood of a cooperative hub ($q_{C|C}$ for the hub) throughout evolutionary process in **g**, and $q_{C|C} - q_{C|D}$ for nodes with different degrees in **h**. Numerical calculations confirm the mechanism we present in **a–c**. Here, we use the same network parameters as Fig. 2. Source data are provided as a Source Data file.

smaller than the average degree $\langle k \rangle$ for a wide range of heterogeneous networks ($C^* < \langle k \rangle$). In contrast, when $\lambda_i = k_i$, the hubs update frequently and $Q < 1$ (Supplementary Note 2), indicating that on average, the number of cooperative neighbours of a cooperator exceeds that of a defector by less than one (Fig. 4c). This leads to a larger critical ratio ($C^* > \langle k \rangle$) for promoting cooperation compared to the scenario with identical update rates shown in Fig. 4a.

We have numerically confirmed the above mechanism on larger scale-free networks. Figure 4d–f show the state of the hub, and the fraction of cooperators among the hub's neighbours over the course of the game dynamics. For $\lambda_i = 1/k_i$, we observe long-lasting periods of cooperation on the hub (Fig. 4e), with infrequent strategy switches from cooperation to defection, which results in the highest $q_{C|C}$ for the hub (Fig. 4g) and in turn the highest $q_{C|C} - q_{C|D}$ over all nodes with different degrees compared to other settings (Fig. 4h). In contrast, fast-updating hubs ($\lambda_i = k_i$) have the lowest average fraction of cooperators among their neighbours (Fig. 4g), leading to a low fraction of cooperative neighbours for the cooperators relative to defectors over the whole network (Fig. 4h). This confirms that degree-inverse update rates promote cooperation on heterogeneous networks because a hub with a low update rate is more conducive to driving its neighbours to cooperation, which further enhances the local dispersal of cooperation among nodes with different degrees.

Furthermore, we find that infrequent updates of hubs can also bring long-term advantages to individuals. We could even consider the general evolutionary process with mutation, where a mutant appears with probability $u$ when the population reaches full cooperation or full defection. Each individual accumulates long-term payoffs during a long period of time. Even with a high mutation rate ($u = 1$), we show that the inverse relationship between update rates and nodes' degrees results in a higher long-term payoff for individuals than identical rates ($\lambda_i = 1$) (Fig. 5a). In contrast, frequent updates of hubs ($\lambda_i = k_i$) lead to a lower payoff than the identical settings (Fig. 5a). This result is also robust over different mutation rates and selection intensities (Fig. 5b, c). Moreover, when the mutation is rare, the population is almost always in full cooperation or full defection, and the time spent in full cooperation (defection) is proportional to $\rho_C$ ($\rho_D$)[50]. Therefore, the settings of update rates which promote cooperation further lead to a higher long-term average payoff, since individuals get $b - c$ in full cooperation but 0 in full defection.

## Theoretical analyses

In addition to the role of hubs that we uncover for the three specific update rate settings ($\lambda_i = 1/k_i, \lambda_i = 1, \lambda_i = k_i$), can we derive the general rule for promoting cooperation that also applies to other distributions of update rates? We next explore how different distributions of $\lambda_i$

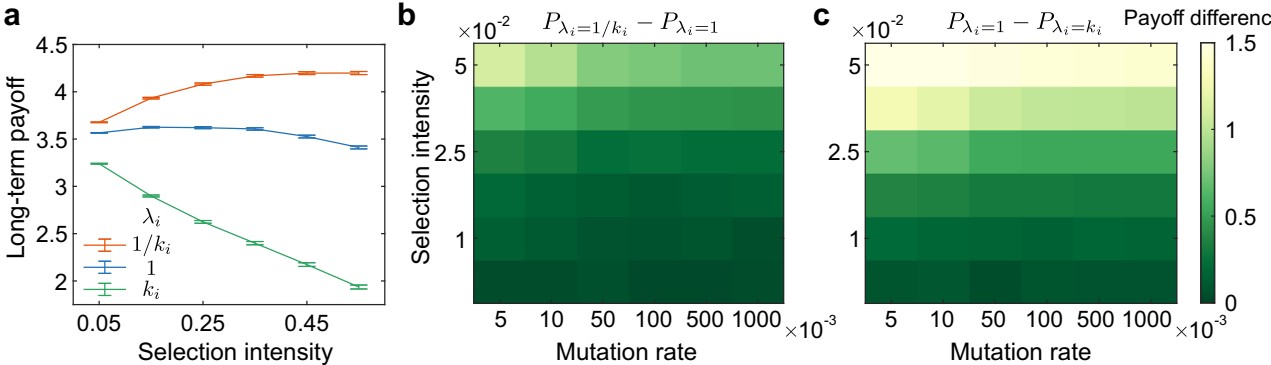

**Fig. 5 | The advantage of infrequent strategy updates of hubs on individual long-term payoffs. a** We calculate the average long-term individual payoff $P_{\lambda_i=1/k_i}, P_{\lambda_i=1}$ and $P_{\lambda_i=k_i}$ corresponding to different update rates ($\lambda_i = 1/k_i, \lambda_i = 1, \lambda_i = k_i$) for 50 individuals during the evolutionary process with mutation rate $u = 1$ on a scale-free network, which are presented as mean values +/- SD. **b** We present the payoff difference $P_{\lambda_i=1/k_i} - P_{\lambda_i=1}$ between $\lambda_i = 1/k_i$ and $\lambda_i = 1$ over different selection intensities and mutation rates. Analogously, the long-term payoff difference between identical rates and rates proportional to nodes' degrees $P_{\lambda_i=1} - P_{\lambda_i=k_i}$ is shown in **c**. The long-term payoff is averaged over $10^3$ independent samples, where the average payoff in each run is obtained over $10^6$ rounds. Source data are provided as a Source Data file.

affect $C^*$ over five different synthetic networks: lattice, random regular, Erdös-Rényi, small-world, and scale-free. For a given network structure, we theoretically predict $C^*$ via equation (1) for uniform, normal, exponential and power-law distributions of the update rate. We find that the critical threshold of a typical homogeneous network—such as a lattice or random regular network—is almost unaffected by the choice of update rate distribution (Fig. 6a, Supplementary Fig. 4). In contrast, heterogeneous structures are quite sensitive, with scale-free networks presenting the most drastic variations in $C^*$ among the different update-rate distributions we consider. This malleability of $C^*$ in heterogeneous networks suggests the possibility of deliberately tuning the update rates to lower the barrier for the emergence of cooperation in a particular network. But to put this into practice, we must first overcome a computational hurdle.

In order to calculate $C^*$ using equation (1), one needs to solve a system of $N(N-1)/2$ linear equations for the recurrence relations between the $\eta_{ij}$ (equation (7) in Methods). Unfortunately, this requires an overall complexity of $\mathcal{O}(N^6)$, rendering the problem intractable for large networks. To circumvent this, we offer an efficient approximation $C^*$ as

$$C^* \approx \frac{N\langle k \rangle^2 \zeta / \langle k^2 \rangle - 1 + \Delta_{\lambda^{(1)}} + \Delta_{\widetilde{\eta_n}}}{N\langle k \rangle \zeta / \langle k^2 \rangle - 1 + \Delta_{\lambda^{(2)}} + \Delta_{\widetilde{\eta_d}}}. \quad (3)$$

This expression obviates the need to solve large systems of linear equations and reduces the computational complexity to $\mathcal{O}(N^3)$. Here $\langle k^2 \rangle$ is the second moment of the degree distribution. We have $\zeta = \sum_{i,j} \frac{k_i k_j \Lambda}{NK^2(\lambda_i + \lambda_j)}$, where $\Lambda = \sum_i \lambda_i$ defines the total rate of update events and $K = \sum_i k_i$ is the summation of all nodes' degrees. Finally, $\Delta_{\lambda^{(1)}}, \Delta_{\lambda^{(2)}}, \Delta_{\widetilde{\eta_n}}$ and $\Delta_{\widetilde{\eta_d}}$ are constants related to the heterogeneity of update rates and coalescence times, the expressions for which are given in Methods. When the update rates are identical, we have $\Delta_{\lambda^{(1)}} = \Delta_{\lambda^{(2)}} = \Delta_{\widetilde{\eta_n}} = \Delta_{\widetilde{\eta_d}} = 0$, and equation (3) recovers the previous results[4,31].

Figure 6b compares the value of $C^*$ predicted by the approximation in equation (3) with that of numerical simulation on two empirical social networks[51,52]. We see that our approximation is remarkably accurate in both networks, regardless of the distribution of the update rates. Moreover, equation (3) offers intuition behind our previous observation that homogeneous structures are robust to different update rates (Fig. 6a). The high symmetry present in these networks means that heterogeneous update rates affect only a limited number

of nodes. For such networks, we have $\Delta_{\widetilde{\eta_n}} \approx \Delta_{\widetilde{\eta_d}} \approx 0$, meaning that $C^* \to \langle k \rangle$ in the limit of large $N$. This coincides with the classical result[4] ($C^* = \langle k \rangle$) regardless of the distribution of update rates.

## A simple condition for the emergence of cooperation

Starting from equation (3) (see Methods), we have the critical benefit-to-cost ratio for large heterogeneous networks

$$C^* \approx \langle k \rangle + \frac{\langle k \rangle^2 \langle k^2 \rangle \Delta_{\widetilde{\eta}(\infty)}}{\langle k \rangle^3 \zeta + (\langle k \rangle^3 - \langle k \rangle \langle k^2 \rangle - \langle k^2 \rangle) \Delta_{\widetilde{\eta}(\infty)}}, \quad (4)$$

where $\langle k \rangle$ is the average degree and $\Delta_{\widetilde{\eta}(\infty)} \approx \frac{\bar{\eta}}{k^2} \sum_{i<j}(k_i - k_j)(\lambda_i - \lambda_j) e_{ij}/(\lambda_i + \lambda_j)$. Note that $\Delta_{\widetilde{\eta}(\infty)} < 0$ when any pair of nodes $i$ and $j$ satisfies the rule $(k_i - k_j)(\lambda_i - \lambda_j) < 0$. When the update rates are identical, we have $\Delta_{\widetilde{\eta}(\infty)} = 0$ and hence $C^* \approx \langle k \rangle$ as expected. In contrast, $C^*$ is smaller (larger) than $\langle k \rangle$ when $\Delta_{\widetilde{\eta}(\infty)} < 0$ ($\Delta_{\widetilde{\eta}(\infty)} > 0$) (Supplementary Note 3.3). Table 1 summarises the values of $C^*$ predicted by equation (4) for the combinations of network structure/update-rate settings.

Taken together, we have theoretically motivated an efficient rule of thumb for lowering the threshold for the emergence of cooperation on large heterogeneous structures. Put simply, the order of any pair of nodes' update rates (for example, $\lambda_i > \lambda_j$) should be reversed from the order of the nodes' degrees (for example, $k_i < k_j$). That is, the one with larger degree should have smaller update rates and vice versa, as is demonstrated in Fig. 6c. In other words, the hubs in networks should update infrequently compared to their neighbours with fewer connections to promote the formation of cooperative clusters, which is consistent with the underlying mechanisms shown in Figs. 3 and 4. A simple but general realisation of this rule is $\lambda_i = 1/k_i^\gamma$ ($\gamma > 0$) which we study numerically in Fig. 2d for different values of $\gamma$. This rule can achieve a lower critical ratio $C^*$ than identical update rates ($\gamma = 0$) on both synthetic heterogeneous (Fig. 2b) and empirical networks ($\gamma = 1$) (Fig. 6b and Supplementary Table 1 and Supplementary Figs. 6 and 7). Meanwhile, the contrary configuration of $\lambda_i = k_i^\gamma$ leads to increases in $C^*$ on heterogeneous networks (Figs. 2c and 6b and Supplementary Figs. 8 and 9).

Moreover, we show that our conclusion can also be applied to other social dilemmas (Supplementary Note 4). For the general two-player game, a cooperator receives rewards $R$ from mutual cooperation, while defectors obtain punishment $P$ from mutual defection. A

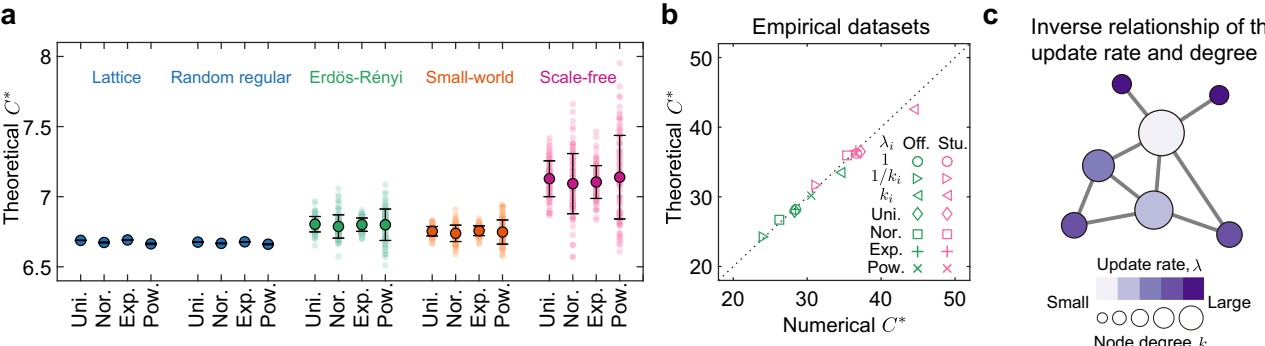

**Fig. 6 | The simple condition for promoting cooperation. a** Illustration of $C^*$ above which cooperation is favoured under uniform (Uni.), normal (Nor.), exponential (Exp.) and power-law (Pow.) distributions of update rates on different structures of networks. Here each dot corresponds to a sample, and the error bars are plotted over 100 samples, indicating the mean values with +/- SD. The robustness of our results with different average degrees is shown in Supplementary Fig. 4. Note that the coupling of node degree and update rate will not bring quantitatively different results on the average $C^*$. The consistent theoretical evidences and details

are given in Supplementary Fig. 5. **b** $C^*$ obtained from our theory (equation (3)) with various update rate ($\lambda_i$) configurations (different markers) are well-matched with the numerical simulations on empirical networks corresponding to face-to-face contacts in an office building[51] (Off.) and a high school[52] (Stu.). Based on the analytical condition given in equation (4), we seek to reduce $C^*$ on large heterogeneous structures, specifically by letting nodes' update rates vary inversely to their degree $k_i$ as shown in **c**, where the size (colour) of nodes captures the magnitude of $k_i$ ($\lambda_i$). Source data are provided as a Source Data file.

defector attempting to exploit a cooperator obtains $T$ and leaves $S$ to its opponent cooperator. We show that cooperation is favoured over defection when $R > P + (T - S)(C^* - 1)/(C^* + 1)$, where a lower threshold for $R$ can be achieved with a lower $C^*$. Note that here $C^*$ is exactly the critical threshold under the donation game. This indicates our conclusion applies to other social dilemmas, such as the general prisoner's dilemma ($T > R > P > S$)[16], snowdrift game ($T > R > S > P$)[3] and stag hunt game ($R > T \geq P > S$)[53].

## The optimal update rate on any network

As an engineering application of designing unmanned and autonomous systems, can we adopt the simple heuristic to favour collective cooperation among agents? Specifically, can we find the optimal set of $\lambda_i$ for a given networked system? To answer this question, we develop OptUpRat, an optimisation algorithm, to search for a set of $\lambda_i$ that minimises $C^*$ (See Box 1, Supplementary Note 5 and Supplementary Fig. 10). Our algorithm OptUpRat is based on RMSProp (root mean square propagation), which is an optimisation algorithm designed for training neural networks[54]. Note that the settings of the learning rate $\epsilon$, decay rate $\rho$ and constant $\delta_{opt}$ parameters are the same as those in RMSprop—the learning rate $\epsilon$ controls the step size of the iteration; $\rho$ controls the decay rate of the moving average; and $\delta_{opt}$ is a small constant added to the denominator to prevent division by zero (see the values of those parameters in Methods). To transform the constrained optimisation with $\lambda_i > 0$ for each individual $i$ into an unconstrained optimisation problem, we define $\lambda_i = \exp(\theta_i)$ to establish a

function mapping from $\theta_i$ to $C^*$. Then the optimal update rate and the corresponding $C^*$ can be obtained via iterative gradient descent, where the gradient is computed by solving a system of $N(N-1)/2$ linear equations after taking the derivative with respect to $\theta_i$ on both sides of equation (7) in Methods.

Consistent with our rule, Fig. 7a shows the scale-free network is more flexible and attain a much smaller threshold at its optimal rate than the lattice. Moreover, the update rates of higher-degree nodes tend to decrease during the optimisation process, while those of smaller-degree nodes increase (Fig. 7b and Supplementary Fig. 11). Interestingly, we find that even on homogeneous structures such like lattices, a policy of identical update rates is not the best choice for promoting cooperation. Indeed, the final update rates deviate significantly from the initial conditions (Fig. 7c and Supplementary Fig. 12). Figure 7d shows that the optimal update rates for different network structures are consistent with our rules shown in Fig. 6c— namely that a node $i$'s update rate $\lambda_i$ should vary inversely with its degree $k_i$.

## Discussion

Our findings reconcile the past conflicting results on how heterogeneous networks affect the evolution of cooperation. Studies that initialise evolutionary game dynamics with an equal number of cooperators and defectors have found that scale-free networks actually outperform homogeneous networks in promoting the evolution of cooperation, as measured by the average fraction of cooperators[6]. But from the perspective of fixation probability, heterogeneous structures impose a higher benefit-to-cost threshold for a single cooperator to take over a population of defectors, at least when all update rates are identical[4,5,31]. This predicts that heterogeneous network structures, despite their ubiquity in physical and social systems, tend to hinder the emergence of collective behaviour. By relaxing this assumption and allowing nodes to update their strategies at non-identical rates, we have shown that scale-free networks can in fact facilitate the fixation of cooperation. As such, degree-heterogeneous networks orchestrated by personalised update rates can be unambiguously conducive to cooperation, provided they are doubly heterogeneous—that is, also heterogeneous in update rate. Taken together, we argue that personalised interaction dynamics and network structure combine to shape the collective dynamics.

From the perspective of microscopic mechanism, we unveil that different update rules render the conflict results. Regarding the frequency of cooperators, previous canonical framework and update rule

## Table 1 | Critical benefit-to-cost ratio $C^*$ for the fixation of cooperation under different update rates and network structures

| Network | Strategy update rate ($\lambda_i$) | Critical ratio ($C^*$) |
|---|---|---|
| Homogeneous | Identical ($\lambda_i = 1$) or heterogeneous | $\approx \langle k \rangle$ |
| Heterogeneous | Identical ($\lambda_i = 1$) | $\approx \langle k \rangle$ |
| | Heterogeneous, $(k_i - k_j)(\lambda_i - \lambda_j) > 0$ | $> \langle k \rangle$ |
| | Heterogeneous, $(k_i - k_j)(\lambda_i - \lambda_j) < 0$ | $< \langle k \rangle$ |

For homogeneous networks, $C^*$ is always equal to the average degree $\langle k \rangle$, irrespective of identical and heterogeneous update rates (Fig. 6a for numerical calculations). While heterogeneous networks can present quantitatively different values of $C^*$ under different update rates (equation (4)), being determined by the relationship between $k_i$ and $k_j$, $\lambda_i$ and $\lambda_j$ of any pair of nodes $i$ and $j$ (Fig. 2b and 2c and 6c). $\lambda_i$ is the update rate for individual $i$ with the number of neighbours $k_i$.

## BOX 1:

# optimisation algorithm OptUpRat

**Input:** Adjacent matrix $E$ of any network

**Output:** the optimal rate $\lambda_i$ for each $i$ and the corresponding critical ratio $C^*$

1. Define $\boldsymbol{\theta} = (\theta_1, \theta_2, ..., \theta_N)^T$, the update rate $\lambda_i = \exp(\theta_i)$

2. Initialise $\boldsymbol{\theta} = \boldsymbol{0}$, learning rate $\epsilon = 1$, decay rate $\rho = 0.9$, constant $\delta_{opt} = 10^{-6}$, squared gradients $\boldsymbol{r} = \boldsymbol{0}$

3. $\mathbf{k_i} \leftarrow \sum_j \mathbf{E_{(i,j)}}, \mathbf{p_{ij}} \leftarrow \mathbf{E_{(i,j)}}/\mathbf{k_i}, \mathbf{p_{ij}^{(n)}} \leftarrow \sum_k \mathbf{p_{ik}^{(n-1)}} \mathbf{p_{kj}}$ for $n = 2, 3$ and any $i, j$

4. Compute $\frac{\partial \mathbf{C}^*}{\partial \eta_{jk}}$ for all $j, k$ according to equation (1)

5. **while** $\frac{1}{N} \sum_i |\Delta\theta_i| > 10^{-6}$

6. $\lambda_i \leftarrow \exp(\theta_i)$ for all $i$

7. Compute $\eta_{ij}$ $(i \neq j)$ by solving the linear system in equation (7)

8. $\eta_{ii} \leftarrow 0$ for all $i$

9. $\mathbf{C}^* \leftarrow \frac{\sum_{i,j} \mathbf{k_i p_{ij}^{(2)}} \eta_{ij}}{\sum_{i,j} \mathbf{k_i p_{ij}^{(3)}} \eta_{ij} - \sum_{i,j} \mathbf{k_i p_{ij}} \eta_{ij}}$ according to equation (1)

10. **for** $i \leftarrow 1$ to $N$

11. Take the derivative with respect to $\lambda_i$ on both sides of equation (7)

12. Compute $\frac{\partial \eta_{jk}}{\partial \lambda_i}$ $(j \neq k)$ by solving the system of $N(N-1)/2$ linear equations

13. $\frac{\partial \eta_{jj}}{\partial \lambda_i} \leftarrow 0$ for all $j$

14. $\frac{\partial \lambda_i}{\partial \theta_i} \leftarrow \exp(\theta_i)$

15. $\frac{\partial \mathbf{C}^*}{\partial \theta_i} \leftarrow \sum_{j,k} \frac{\partial \mathbf{C}^*}{\partial \eta_{jk}} \frac{\partial \eta_{jk}}{\partial \lambda_i} \frac{\partial \lambda_i}{\partial \theta_i}$

16. **end for**

17. $\boldsymbol{g} \leftarrow \left(\frac{\partial \mathbf{C}^*}{\partial \theta_1}, \frac{\partial \mathbf{C}^*}{\partial \theta_2}, ..., \frac{\partial \mathbf{C}^*}{\partial \theta_N}\right)^T$

18. $\boldsymbol{r} \leftarrow \rho\boldsymbol{r} + (1-\rho)\boldsymbol{g} \odot \boldsymbol{g}$

19. $\boldsymbol{\theta} \leftarrow \boldsymbol{\theta} - \frac{\epsilon}{\sqrt{\delta_{opt} + \boldsymbol{r}}} \odot \boldsymbol{g}$

20. **end while**

21. **return** $\lambda_i, C^*$

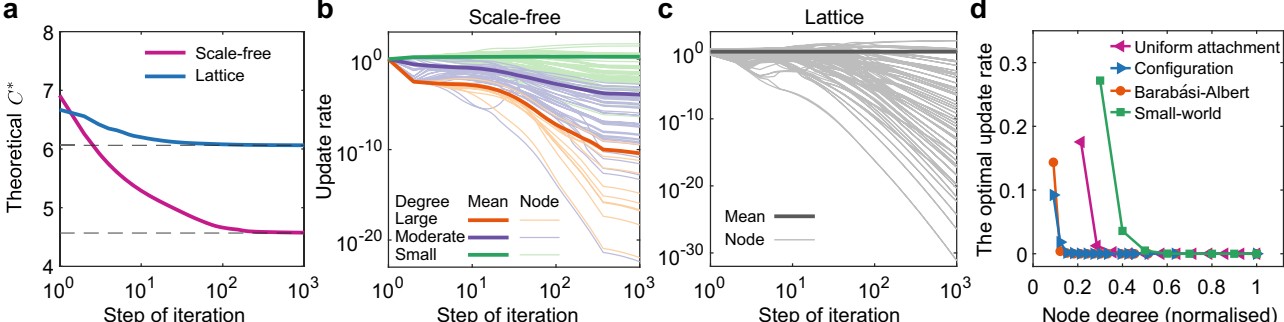

**Fig. 7 | Designing the optimal update rates to promote cooperation on heterogeneous networks. a** We present the convergence of the objective function $C^*$ for a scale-free network (purple) and a lattice (blue) over $10^3$ iterations of our optimisation algorithm OptUpRat. **b** The corresponding evolution of the (tunable) $\lambda_i$ for all nodes, which are divided into three categories (large, moderate and small), based on the range of degrees in the scale-free network. The mean update rate among individuals in each category is shown with the thicker line. We see that the optimal update rates tend to decrease for large nodes (orange) and generally increase for small nodes (green). **c** For the lattice, the optimal update rate also presents the deviations from the identical rate. Beyond presenting the detailed process for optimising $C^*$ in panels **a**–**c** we show the final $\lambda_i$ compared to the nodes' degree for scale-free networks (generated by the configuration model[67], Barabéasi-Albert model[32]), small-world network[66] (rewiring probability 0.7) and networks constructed from a uniform attachment model[68] in **d**, where we normalise the optimal update rate and the node degree. We again observe an inverse relationship between the final update rates and the corresponding nodes' degree, consistent with our rule shown in Fig. 6c. Here we use the same network parameters as Fig. 2. Source data are provided as a Source Data file.

naturally lead to infrequent strategy switching (Supplementary Note 6)[6,33,34]. This facilitates the formation of cooperative clusters and leads to a high fraction of cooperators on heterogeneous networks. Previous findings are consistent with the underlying microscopic mechanism in our study, namely infrequent updates of hubs facilitate the emergence of cooperation. Indeed, by applying the canonical death-birth update with identical rates in the framework analysing the frequency of cooperators[6,33,34], we find that heterogeneous networks impede the average frequency of cooperators compared to homogeneous scenarios (Supplementary Fig. 13).

Furthermore, we compare our results with experimental studies on cooperation in heterogeneous networks. Consistent with our

theoretical findings, there is an insightful experimental study also reporting that heterogeneous networks do not promote cooperation in prisoner's dilemmas[55]. In this behavioural experiment, a player's decisions to cooperate or defect are relevant to the level of cooperation in their neighbourhoods, which renders the network irrelevant. Therefore, the main difference between this experimental finding and our study lies in the update rules. Specifically, players are more likely to imitate the strategy from neighbours with higher payoffs in our theoretical framework. To further uncover the behavioural dynamics from the perspective of fixation probability, a promising future application involves the design of human behavioural experiments starting from a single cooperator and ending with full cooperation or defection. Comparing the individual behavioural mode in experiments from these two perspectives will facilitate the understanding of the emergence of cooperation in realistic scenarios.

A natural extension of our findings is exploring the scenario with multiple strategies[56–58]. In this way, the diverse strategy update rhythms may couple multiple strategies with complex dynamics. In addition, our findings may contribute to the study of network formation, elucidating the factors influencing group formation, such as individuals' propensity to establish connections with those who share similar rhythms. Specifically, discovering the scenarios wherein individuals with similar update rates are allowed to construct a group may provide valuable information regarding the optimal network configuration in the context of heterogeneity.

One promising direction for future research lies in evolutionary dynamics on temporal networks. Time-varying network structure is a recurring theme in social systems, encoding not only who interacts with whom but with when (and how often) these interactions happen[59]. It was recently discovered that temporal networks generally enhance the evolution of cooperation relative to comparable static networks[12], yet the practical scenarios easily trigger the heterogeneous time rhythm of strategy updating. In real temporal networks, a node's degree may vary drastically even over short time periods[47,48,60]. This—in tandem with other temporal effects such as burstiness and multi-frequency interactions[47,61]—may lead to more exotic evolutionary dynamics. By regarding a temporal network as a sequence of static snapshots, our theory might be adopted to further tailor individuals' update rates in temporal evolutionary game dynamics.

## Methods

### Evolutionary process

In each round of the game, individuals interact with their neighbours and accumulate the payoffs accordingly. The payoff matrix of the game is given by

$$
\begin{array}{cc}
 & \begin{array}{cc} C & \quad D \end{array} \\
\begin{array}{c} C \\ D \end{array} & \begin{pmatrix} b-c & -c \\ b & 0 \end{pmatrix}
\end{array}.
$$

The state of network at any given time can be encoded by a binary vector $\mathbf{x} \in \{0,1\}^N$, where $x_i = 1$ denotes that the player $i$ chooses strategy C, otherwise $x_i = 0$ indicates strategy D. Using this representation of the network state $\mathbf{x}$, $i$'s average payoff is $f_i(\mathbf{x}) = -cx_i + b\sum_j p_{ij}x_j$, where $p_{ij} = e_{ij}/k_i$ indicates the probability of a single step random walk from $i$ to $j$ on the network. For a node $i$ with update rate $\lambda_i$, the probability to be chosen for a strategy update is $\lambda_i/\Lambda$, where $\Lambda = \sum_i \lambda_i$ defines the total rate of update events. It follows that at the end of each round, the probability for a player $j$ to transmit its strategy to $i$ is $r_{ji}(\mathbf{x}) = \frac{\lambda_i}{\Lambda} \frac{e_{ij}F_j(\mathbf{x})}{\sum_l e_{il}F_l(\mathbf{x})}$, where $F_j(\mathbf{x}) = 1 + \delta f_j(\mathbf{x})$ indicates the fitness of individual $j$. Note that the fixation probability does not change when the rate of strategy updates

for each individual is identical since the normalised update rates are the same.

### Fixation probability

As shown in the Supplementary Note 1, the fixation probability of cooperation is derived by a first-order expression as the neutral fixation probability ($1/N$) plus a correction term due to weak selection, namely

$$
\rho_C = \frac{1}{N} + \delta \left\langle \frac{d}{d\delta} \Big|_{\delta=0} \widehat{\Delta}(\mathbf{x}) \right\rangle_{\mathrm{u}}^{\circ} + O(\delta^2), \tag{5}
$$

where $\widehat{\Delta}(\mathbf{x})$ denotes the reproductive-value-weighted frequency change of cooperation, which is given by

$$
\widehat{\Delta}(\mathbf{x}) = \sum_i \frac{k_i}{\lambda_i \sum_l \frac{k_i}{\lambda_l}} \sum_j (x_j - x_i) r_{ji}(\mathbf{x}). \tag{6}
$$

Here $\langle \varphi \rangle_{\mathrm{u}}^{\circ}$ indicates the summation of the expectation of $\varphi$ with $\varphi(\mathbf{1}) = \varphi(\mathbf{0}) = 0$ under neutral drift through time step $t = 0$ to infinity, namely $\langle \varphi(\mathbf{x}) \rangle_{\mathrm{u}}^{\circ} = \sum_{t=0}^{\infty} \sum_{\mathbf{x} \in \{0,1\}^N} \mathbb{P}_{\mathrm{u}}^{\circ}[\mathbf{X}(t) = \mathbf{x}] \varphi(\mathbf{x})$, where $\mathbb{P}_{\mathrm{u}}^{\circ}[\mathbf{X}(t) = \mathbf{x}]$ indicates the neutral probability of the system reaching state $\mathbf{x}$ at time step $t$ starting from the initial state with a single uniformly selected cooperator in population with $N-1$ defectors. Combining equations (5) and (6), the fixation probability can be expressed as

$$
\rho_C = \frac{1}{N} + \frac{\delta}{\Lambda \sum_i \frac{k_i}{\lambda_i}} \left[ -c \sum_{i,j} k_i p_{ij}^{(2)} \eta_{ij} + b \left( \sum_{i,j} k_i p_{ij}^{(3)} \eta_{ij} - \sum_{i,j} k_i p_{ij} \eta_{ij} \right) \right] + O(\delta^2),
$$

where $\eta_{ij} = \left\langle \hat{x} - x_i x_j \right\rangle_{\mathrm{u}}^{\circ}$, and $\hat{x} = \sum_i \pi_i x_i$ represents the reproductive-value-weighted frequency of cooperators, where $\pi_i$ is the reproductive value[62–64] uniquely solved by Supplementary equation (2), quantifying the expected contribution of site $i$ to the future gene pool under neutral drift. Here $\eta_{ij}$ satisfies the recurrence relation of

$$
\eta_{ij} = \begin{cases} \frac{\Lambda}{N(\lambda_i + \lambda_j)} + \sum_k \frac{\lambda_i}{\lambda_i + \lambda_j} p_{ik} \eta_{kj} + \sum_k \frac{\lambda_j}{\lambda_i + \lambda_j} p_{jk} \eta_{ki}, & i \neq j \\ 0, & i = j \end{cases}. \tag{7}
$$

By letting $\rho_C > 1/N$, we obtain $C^*$ shown in equation (1).

### Calculation of the critical ratio $C^*$

We first define $\eta^{(n)} = \sum_{i,j} k_i p_{ij}^{(n)} \eta_{ij}/K$, where $K = \sum_i k_i$ is the summation of all nodes' degrees, then equation (1) can be rewritten as

$$
C^* = \frac{\eta^{(2)}}{\eta^{(3)} - \eta^{(1)}}.
$$

From the recurrence relation of $\eta_{ij}$ in equation (7), we further derive the recurrence relation of $\eta^{(n)}$ with

$$
\eta^{(n)} = \sum_{i,j} \frac{k_i}{K} p_{ij}^{(n)} \frac{\Lambda}{N(\lambda_i + \lambda_j)} + \widetilde{\eta}^{(n+1)} - \sum_i \frac{k_i}{K} p_{ii}^{(n)} \eta_{ii}^+, \tag{8}
$$

where $\widetilde{\eta}^{(n+1)} = \sum_{i,j,l} \frac{k_i}{K} p_{ij}^{(n)} \frac{2\lambda_j}{\lambda_i + \lambda_j} p_{jl} \eta_{il}$ and $\eta_{ii}^+ = \frac{\Lambda}{2N\lambda_i} + \sum_l p_{il} \eta_{il}$.

By defining the difference $\Delta_{\widetilde{\eta}^{(n)}} := \widetilde{\eta}^{(n)} - \eta^{(n)}$ and using the recurrence relation of equation (8), we obtain the calculation of $C^*$ shown in equation (3) with mean-field approximation, with $\Delta_{\widetilde{\eta}_n} = -\Delta_{\widetilde{\eta}^{(2)}} + \frac{K^2}{\sum_i k_i^2} \Delta_{\widetilde{\eta}^{(\infty)}}$ and $\Delta_{\widetilde{\eta}_d} = -\Delta_{\widetilde{\eta}^{(2)}} - \Delta_{\widetilde{\eta}^{(3)}} + \frac{KN}{\sum_i k_i^2} \Delta_{\widetilde{\eta}^{(\infty)}}$ for simplification, where $\Delta_{\lambda^{(1)}} = \sum_i \frac{k_i}{2K} \left( 1 - \frac{\Lambda}{N\lambda_i} \right) + \sum_{i,j} \frac{k_i}{2K} p_{ij} \left( 1 - \frac{2\Lambda}{N(\lambda_i + \lambda_j)} \right)$ and $\Delta_{\lambda^{(2)}} = \sum_{i,j} \frac{k_i}{2K}$

$(p_{ij} + p_{ij}^{(2)})(1 - \frac{2\Lambda}{N(\lambda_i + \lambda_j)})$. According to Supplementary Note 3, we further have $\Delta_{\tilde{\eta}^{(2)}} \approx N\Delta_{\tilde{\eta}^{(\infty)}}/\langle k \rangle$ and $\Delta_{\tilde{\eta}^{(3)}} \approx N\Delta_{\tilde{\eta}^{(\infty)}}/\langle k \rangle^2$ for large networks, and hence $C^*$ shown in equation (4) follows immediately.

## Reporting summary

Further information on research design is available in the Nature Portfolio Reporting Summary linked to this article.

## Data availability

Source data are provided as a Source Data file. Data of empirical networks analysed in Fig. 6b are publicly available and can be found in the corresponding references[51,52]. Source data are provided with this paper.

## Code availability

The codes are written using MathWorks MATLAB R2021a and Python 3.8.5. All source codes related to the work can be found at[69] https://github.com/yaomeng1/PersonalizedStrategyUpdates.

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

## Acknowledgements

Y.M. and A.L. are supported by National Key Research and Development Program of China (2022YFA1008400), National Natural Science Foundation of China (62173004), Beijing Nova Program (Z211100002121105) and SMP-IDATA Chenxing Youth Fund. Y.-Y.L. is supported by National Institutes of Health (R01AI141529, R01HD093761, RF1AG067744, UH3OD023268, U19AI095219, and U01HL089856).

## Author contributions

Y.M. and A.L. conceived, designed, and performed the research. All authors analysed the results. Y.M. performed mathematical calculations and numerical simulations under the direction of A.L. A.L., S.P.C. and Y.M. wrote the manuscript, and Y.-Y.L. edited the manuscript.

## Competing interests

The authors declare no competing interests.
