## [Peer Review File · Nature Communications]

Dynamics of collective cooperation under personalised strategy updatesReviewers' Comments:

Reviewer #1:

Remarks to the Author:

In the paper on the Evolution of collective cooperation under arbitrary strategy updates the authors have analysed an evolutionary process in a structured population with heterogeneous strategy updating rates. The authors focus on relaxing the assumptions of homogeneity in two dimensions and present a comprehensive set of results that convincingly suggest that heterogeneity in updating rates might be beneficial for the evolution of cooperation. I found this paper very clearly written and, as I was reading the manuscript, the authors provided answers to the questions I had, while also suggesting interesting extensions beyond my expectations. I believe this work provides a significant contribution to our understanding of how individual inequality affects the evolution of cooperation.

While being overall quite positive about the manuscript, I have several comments that might help the authors to improve the presentation of their results. Specifically:

1. My first and main comment relates to the presented numerical results and the absence of sensitivity analysis wrt to the parameter values. That is, in all figures the results are present for a fixed set of parameters, which is fine, however, the effect of changes in those parameters is neither presented nor discussed. I think that an additional set of simulations for a wider range of N , k and Δ in spirit of simulations presented in Figure 2 would be beneficial for the reader's appreciation and understanding of the results.

2. While I understand that analytical results cannot be obtained for higher selection strength, I still wonder if the authors can say something about the behaviour of the process for values of Δ much higher than ϵ ? Say, 10, 100, 1000? I would be very curious to see a couple of simulations related to this point.

3. The authors provide a comprehensive set of analytical results showing that if players have equal update rates (independent of the exact value), the critical C^* should be equal to k , where k is the average number of neighbours. However, I wonder if this holds up in numerical simulations if we set $\lambda=1/k$ or $\lambda=k$ for all players (in comparison to $\lambda=1$ chosen by the authors). That is, I am curious if the speed of update has a more general effect in addition to heterogeneity?

4. I found the simple rule of thumb in lines 202-204 fascinating. However, I feel that for a broader audience the authors might want to spend a bit more time here explaining what does this mean for the network, for the process, and for the evolution. Perhaps, connections to some examples or applications might be useful here.

In addition to this, I wonder if the authors can say something more about the existence of a threshold on either k_i or k for a fixed λ ?

5. In the future directions, I wonder if the authors can connect their results and model setup to a broader literature on more sophisticated strategies (non-binary C or D players).

In addition, their results might say something interesting in relation to more general network formation literature, and why people form groups in particular (e.g. people are more likely to connect to the individuals with similar rhythms). Perhaps, fixing a set of players with λ_i and letting them to form a network would tell us something about the optimal network in the presence of heterogeneity.

Minor comments:

1. I believe that the general statement in lines 31-33 is only true in the absence of other mechanisms (such as direct, indirect or network reciprocity, punishment, etc). Some references to a broader set of literature would be very much appreciated.

2. I think the section on "Theoretical analyses" jumps into technical details a bit abruptly. Perhaps, one has to motivate a bit more for why we would want to consider different distributions of λ and how can they be interpreted (what does it mean for the updating rule). Here, again, the question is how sensitive are the results wrt parameters used in Figure 5?

3. In the algorithm, what are the learning rate, decay rate and constant Δ_{opt} ? how are they

defined, why the values set this way and what is their effect? Perhaps, a more broader discussion of the algorithm in the main text would also be good.

4. In the methods section, I think it would be beneficial to repeat previous notation descriptions (such as Λ , K or $F_j(\mathbf{x})$).

5. What is the reproductive-value-weighted frequency \hat{x} ?

Comments on the figures:

1. Figure 1: what are the update rates λ_i chosen for each player on the right?

2. Figure 2d: could you please add a row for $\lambda=k_i$?

3. Figure 2e: I found it very hard to read the markers. Similar comment about Figure 5b.

4. Figure 5a: while there is certainly more variability in the results for ER, SM and SF networks, the average C^* are still kind of similar across different distributions. Can the authors comment on that?

Math notation comments:

1. I wonder if \mathcal{C}^* was used before or if the authors come up with this for the first time? Mathematically, $\mathcal{\{}}$ usually refers to sets, not to values of a variable. But if this is a common notation, let it be.

2. Why in the main text the edges are denoted by e_{ij} while in the algorithm the authors are using the adjacent matrix A and A_{ij} ? are these two different? Also, I personally believe that it is mathematically more elegant to denote vectors as small bold and matrices as capital non-bold letters.

3. Am I correct in understanding that \mathbf{u} in equation (5) is used to denote a uniform draw? Why is it bold?

Reviewer #2:

Remarks to the Author:

By calculating the fixation probability of a single cooperative mutant, the authors demonstrate that dynamical heterogeneity (in the form of differences in strategy update rates) can promote cooperation in structurally heterogeneous networks (e.g., scale-free, small world, or Erdos-Renyi). The authors find that the evolution of cooperation in heterogeneous networks depends crucially on the distribution of learning rates: if rates increase with a node's degree, this hinders the evolution of cooperation; if the reverse is true, cooperation evolves more easily. This result is used to develop an algorithm to find the optimal update rates to promote the emergence of cooperation on a given network.

This study provides new insights on the interplay between dynamical and structural heterogeneity and the evolution of cooperation and has the potential to make an important contribution to the literature on heterogeneity and cooperation. However, there are two major points that the authors need to address:

1) The first point concerns the lack of arguments about the evolution of optimal update rates. The authors demonstrate that certain distributions of update rates (i.e., when update rates decrease with increasing node degree) favour cooperation. But the question remains: why and under what conditions would natural selection promote such distributions of update rates? In other words, is it always advantageous for individual nodes to modulate their update rate so that it is inversely proportional to their degree?

If the answer to the above question is yes, then distributions of learning rate that promote cooperation will evolve. But it is also possible that individual nodes maximise their payoffs by increasing their learning rate proportionally to their number of connections: for example, if a cooperative hub has many cooperative neighbours, it could be advantageous (at least, in the short run) to change strategy to defection in order to exploit them.

If this is indeed the case, natural selection would not promote the evolution of distributions of learning

rates that favour cooperation. The authors' results would be more relevant to understanding the behaviour of agent-based models than the evolution of biological populations. The relevance and robustness of the results would be enhanced by a clear explanation of why, and under what conditions, natural selection would lead to a distribution of learning rates that favours cooperation.

2) The second point concerns the significance of the work in relation to the existing literature. I disagree with the authors' claim that their result (that heterogeneous update rates promote cooperation) is "surprising" and "undercuts the conventional wisdom that heterogeneous structure is generally antagonistic to cooperation". The main study cited to support this claim is Allen et al. (2017). Allen et al.'s find an exact formula that expresses the critical benefit to cost ratio $(b/c)^*$ above which cooperation will evolve as a function of coalescence time. The degree distribution does not figure explicitly in it, and it is not very obvious to me that a more heterogeneous network will automatically lead to a higher $(b/c)^*$.

On the contrary, previous studies directly comparing homogeneous and scale-free networks revealed that a heterogeneous degree distribution promotes cooperation, both in the prisoner's dilemma (Santos and Pacheco, 2005; Santos, Rodrigues, and Pacheco, 2006) and in other social dilemmas (stag hunt and snowdrift; Santos, Pacheco, and Lenaerts, 2006). There are also other studies indicating that heterogeneity (in the form of random payoff fluctuations) is a powerful force in maintaining cooperation in structured populations (see, for example, Javarone and Amaral, 2020).

While the studies above do not explicitly calculate fixation probabilities, but only the average frequency of cooperators, it seems important to clarify why these two approaches lead to different conclusions. The authors acknowledge this difference in the discussion, but to me they do not satisfactorily explain why the difference exists in the first place.

One possible explanation might be that, in the update function used in the Santos and Pacheco (2005) and the two Santos et al. (2006) studies, the probability that node x imitates the strategy of node y is proportional to the ratio between payoff differences and the maximum degree of either node. Therefore, it is easier for a high-payoff strategy to spread between nodes with a low degree than to or from a hub. Is this analogous to one of the conditions studied in this manuscript (lower update rates for hubs)? The authors will certainly have a better sense of whether these two mechanisms are similar or not; perhaps this should be addressed in the discussion.

To summarise: it seems to me that the novelty of this study lies not in the surprising nature of the main results, but rather in its potential to provide a unifying framework to understand the interplay between structural heterogeneity, learning rates, and cooperation. To better integrate their work into the existing literature, the authors should address in more detail why different approaches have yielded different conclusions, acknowledging the existing literature that links heterogeneity with cooperation, and perhaps comparing their results with experimental studies on cooperations in heterogeneous networks (e.g., Gracia-Lázaro et al., 2012).

Finally, the manuscript would benefit from a few minor clarifications:

3) I find the word "arbitrary" potentially confusing in the context of learning rates. It might give the impression that each node could *arbitrarily* update its learning rate (e.g., through some evolutionary dynamics), which is obviously not the case. The learning rates are different from node to node, but fixed for the duration of a simulation.

4) The authors might want to clarify that they are using a simultaneous donation game to model the evolution of cooperation and that this is a specific type of prisoner's dilemma; the two are not equivalent. I'd be interested if the authors have a sense of whether their general conclusions would also apply to more general forms of the prisoner's dilemma and, more broadly, to other social

dilemmas as well.

5) Abstract: the sentence "Combining theoretical and computational techniques with synthetic and empirical data analyses" is potentially misleading, as the paper does not present or analyse any empirical data. In fact, I think it would be very interesting to compare the results of this study with empirical findings on cooperation in heterogeneous networks.

REFERENCES

- Santos, F. C., & Pacheco, J. M. (2005). Scale-free networks provide a unifying framework for the emergence of cooperation. *Physical review letters*, 95(9), 098104.
- Santos, F. C., Rodrigues, J. F., & Pacheco, J. M. (2006). Graph topology plays a determinant role in the evolution of cooperation. *Proceedings of the Royal Society B: Biological Sciences*, 273(1582), 51-55.
- Santos, F. C., Pacheco, J. M., & Lenaerts, T. (2006). Evolutionary dynamics of social dilemmas in structured heterogeneous populations. *Proceedings of the National Academy of Sciences*, 103(9), 3490-3494.
- Amaral, M. A., & Javarone, M. A. (2020). Heterogeneity in evolutionary games: an analysis of the risk perception. *Proceedings of the Royal Society A*, 476(2237), 20200116.
- Gracia-Lázaro, C., Ferrer, A., Ruiz, G., Tarancón, A., Cuesta, J. A., Sánchez, A., & Moreno, Y. (2012). Heterogeneous networks do not promote cooperation when humans play a Prisoner's Dilemma. *Proceedings of the National Academy of Sciences*, 109(32), 12922-12926.

Reviewer #3:

Remarks to the Author:

The paper focuses on the evolution of cooperation in networks. Typically, in network models, the update rate of nodes is fixed or depends on a stochastic process, which is uniform across all nodes. This paper introduces the concept of heterogeneous strategy update rates as a factor that could facilitate the evolution of cooperation. It provides a theoretical framework and derives its results both analytically and through simulations.

The paper is well-written, engaging, and to the point. The clarifications and examples are very helpful for readers. I enjoyed reading the paper.

My main concern regards the relevance and generality of its main results. The paper demonstrates the effect of non-uniform update rates in a very specific setting: when they are inversely proportional to the average connectivity of a node. My suggestion is to address several important questions: Why should we expect update rates to differ, and to what extent? Are randomly distributed update rates (e.g., following a Poisson process) incorrect or insufficient for modeling and approximating the nature of different updating events? If so, why should we expect a negative correlation between connectivity and update rate? I believe the introduction overlooks these crucial questions. The paper could be strengthened by adding real-world relevance to this modification. For example, one might argue that a higher number of connections could slow down the updating process due to more information needing processing. However, the results are intriguing in the opposite case: when update rates vary inversely with the number of connections. This not only seems like a very specific case but also appears to be an unlikely scenario (correct me if I am wrong and my apologies if that's the case). The authors also provide real-world network examples (office, student, Attiro family contacts, and San Juan family contacts) and demonstrate how the critical benefit-to-cost ratio would change based on the

assumption of update rates. Justifying the main result for these networks would be beneficial. Without solid real-world relevance, the paper remains interesting but might be better suited for a specialist journal.

Related to this, the abstract suggests that this update rate extension can be relevant for design purposes. The authors went out of their way to provide an algorithm to optimize collective cooperation by changing individuals' update rates. (Thanks also for the Matlab code.) Although that's very interesting, the same question of behavioral relevance arises. Update rates seem to me a highly endogenous aspect. It's challenging to think of many ways to intervene in the update rates in real-world scenarios (as opposed to changing the network structure, for instance). Elaborating on this would be immensely helpful.

Another aspect is the paper's limited connection to previous studies dealing with update mechanisms and rates. I am not an expert of this particular topic so it might be me. But it's somewhat difficult to ascertain the state of the literature in this area. It would help, especially for interdisciplinary readers, if the paper could situate itself within the context of other studies on heterogeneous update rates, if any exist.

I leave the judgment and decision to the authors, but here are some studies that caught my attention:

Allen, James M., and Rebecca B. Hoyle. 'Asynchronous Updates Can Promote the Evolution of Cooperation on Multiplex Networks'. *Physica A: Statistical Mechanics and Its Applications* 471 (April 2017): 607–19.

Grilo, Carlos, and Luís Correia. 'Effects of Asynchronism on Evolutionary Games'. *Journal of Theoretical Biology* 269, no. 1 (January 2011): 109–22.

———. 'The Influence of the Update Dynamics on the Evolution of Cooperation'. *International Journal of Computational Intelligence Systems* 2, no. 2 (June 2009): 104–14.

Johnson, Tim, and Oleg Smirnov. 'Temporal Assortment of Cooperators in the Spatial Prisoner's Dilemma'. *Communications Biology* 4, no. 1 (12 November 2021): 1283.

Wang, Dongqi, Xuanyue Shuai, Qihui Pan, Jingye Li, Xiaolong Lan, and Mingfeng He. 'Long Deliberation Times Promote Cooperation in the Prisoner's Dilemma Game'. *Physica A: Statistical Mechanics and Its Applications* 537 (January 2020): 122719.

Zhang, Jianlei, Chunyan Zhang, Ming Cao, and Franz J. Weissing. 'Crucial Role of Strategy Updating for Coexistence of Strategies in Interaction Networks'. *Physical Review E* 91, no. 4 (2 April 2015): 042101.

Minor Comments:

- The abstract mentions (line 20): "Combining theoretical and computational techniques with synthetic and empirical data analyses, we find that when individuals' update rates vary inversely with their number of connections, heterogeneous connections actually outperform homogeneous ones in promoting cooperation." If I'm not mistaken, the empirical data analyses do not support or refute the claim but rather demonstrate the application of the logic. So, it might be clearer to specify this, whether I am right or wrong in my understanding.

- I really appreciate the effort the paper puts into providing intuition. Regarding the "lock-in" explanation of the mechanism (line 122), if you could also explain why this effect is not symmetric for cooperation and defection, and whether it favors cooperation, that would assist the reader.

Happy to read the paper and I hope my comments make sense to the authors.

Response to Reviewer #1:

In the paper on the Evolution of collective cooperation under arbitrary strategy updates the authors have analysed an evolutionary process in a structured population with heterogeneous strategy updating rates. The authors focus on relaxing the assumptions of homogeneity in two dimensions and present a comprehensive set of results that convincingly suggest that heterogeneity in updating rates might be beneficial for the evolution of cooperation. I found this paper very clearly written and, as I was reading the manuscript, the authors provided answers to the questions I had, while also suggesting interesting extensions beyond my expectations. I believe this work provides a significant contribution to our understanding of how individual inequality affects the evolution of cooperation.

While being overall quite positive about the manuscript, I have several comments that might help the authors to improve the presentation of their results. Specifically:

We thank Reviewer #1 for reviewing our manuscript and the overall positive assessment. We next address each of the comments in order.

1. My first and main comment relates to the presented numerical results and the absence of sensitivity analysis wrt to the parameter values. That is, in all figures the results are present for a fixed set of parameters, which is fine, however, the effect of changes in those parameters is neither presented nor discussed. I think that an additional set of simulations for a wider range of N , k and δ in spirit of simulations presented in Figure 2 would be beneficial for the reader's appreciation and understanding of the results.

We thank Reviewer #1 for this constructive comment. Motivated by this comment, we conducted extensive simulations using parameters $N = 50, 200$, $k = 4, 8$ and $\delta = 0.025, 0.05, 0.1$ to show the robustness of our results (see Supplementary Figs. 1-3). Moreover, we added "*We show the robustness of our results over different population size, average degree and selection intensity in Supplementary Figs. 1-3.*" to lines 118-120.

2. While I understand that analytical results cannot be obtained for higher selection strength, I still wonder if the authors can say something about the behaviour of the process for values of δ much higher than ϵ ? Say, 10, 100, 1000? I would be very curious to see a couple of simulations related to this point.

We thank Reviewer #1 for this very insightful comment. The simple reason we consider weak selection strength is that, for a strong intensity of selection, cooperation will generally be ruled out in *any* network. Indeed, the role of the selection intensity δ is to map individual payoff f_i to fitness $F_i = 1 + \delta f_i$. For example, when $\delta = 1$, the initial cooperator (with all defecting neighbours) obtains the lowest possible payoff -1 and thus has the lowest fitness of 0. The defecting neighbours will not imitate this initial cooperator, resulting in the immediate extinction of cooperation if the initial cooperator is selected to update. A stronger selection intensity will lead to a worse scenario for the initial cooperator.

Motivated by this comment, we have added "*For strong selection intensity, cooperation is disfavoured since the initial cooperator will not be able to survive or spread its strategy.*" to the main text (lines 90-91).

3. The authors provide a comprehensive set of analytical results showing that if players have equal update rates (independent of the exact value), the critical C^* should be equal to k , where k is the average number of neighbours. However, I wonder if this holds up in numerical simulations if we set $\lambda = 1/k$ or $\lambda = k$ for all players (in comparison to $\lambda = 1$ chosen by the authors). That is, I am curious if the speed of update has a more general effect in addition to heterogeneity?

We thank Reviewer #1 for this constructive comment. The total update rate has no general effect on the fixation probability because it is normalised out by design. In the evolutionary process, the probability that individual i is chosen to update given there is an update event in the system is λ_i/Λ , where Λ is the total rate over all individuals. In this way, each individual i is chosen to update with probability λ_i/Λ after each round of the game

in the numerical simulations. Therefore, the settings of $\lambda_i = 1/\langle k \rangle$, $\lambda_i = \langle k \rangle$ or $\lambda_i = 1$ would lead to the same results.

Motivated by this comment, we have added “*Note that the fixation probability does not change when the rate of strategy updates for each individual is identical since the normalised update rates are the same.*” to the revised manuscript (lines 348-350).

4. I found the simple rule of thumb in lines 202-204 fascinating. However, I feel that for a broader audience the authors might want to spend a bit more time here explaining what does this mean for the network, for the process, and for the evolution. Perhaps, connections to some examples or applications might be useful here.

Following the excellent suggestion of Reviewer #1, we have added the sentence “*In other words, the hubs in networks should update infrequently compared to their neighbours with fewer connections to promote the formation of cooperative clusters, which is consistent with the underlying mechanisms shown in Figs. 3 and 4.*” (see lines 241-244).

In addition to this, I wonder if the authors can say something more about the existence of a threshold on either k_i or k for a fixed λ ?

Sure. When the mean degree $\langle k \rangle$ is small, we have proven that cooperation can be favoured. However, for the largest possible $\langle k \rangle$ (i.e., $\langle k \rangle = N - 1$ in a complete graph), cooperation is disfavoured according to the replicator dynamics [25], validating the existence of the threshold of the mean degree $\langle k \rangle$.

To demonstrate this point intuitively, we take an example by calculating C^* for all networks of size $N = 6$ in Fig. R1a and $N = 7$ in Fig. R1b. We take an update rate sequence of $\{1, 2, \dots, N\}$ for the N nodes. Note that for each network structure, the configuration of update rates includes all possible orderings of nodes. In other words, the sequence of the nodes can correspond to any shuffled order of update rates sequence. We show that when the mean degree is higher than a threshold (vertical dashed line), all network structures with any order of update rate sequence will not favour cooperation, since $C^* < 0$ indicates that the $\rho_c < 1/N$ for any $b > c > 0$ (As shown by the dots in Fig. 4a in ref. [5]). In this way, to facilitate the emergence of cooperation, the mean degree of networks $\langle k \rangle$ should not be too large.

Figure R1: Critical benefit-to-cost ratio, C^* , for all networks of size six and seven. We calculate the critical benefit-to-cost ratio C^* for all networks of size $N = 6$ in **a** and $N = 7$ in **b**. We take the fixed setting of update rate $\{1, 2, 3, 4, 5, 6\}$ and $\{1, 2, 3, 4, 5, 6, 7\}$ for networks of size six and seven. With the fixed λ_i , the network structure and the sequence of nodes can be arbitrary. For networks of size six (seven), we calculate C^* for 112 (853) network structures where each has 720 (5040) possible sequences of update rates. When the mean degree of the network is larger than a threshold (grey dashed line), there is no network structure or settings of update rates favouring the emergence of cooperation.

5. In the future directions, I wonder if the authors can connect their results and model setup to a broader literature on more sophisticated strategies (non-binary C or D players).

In addition, their results might say something interesting in relation to more general network formation literature, and why people form groups in particular (e.g. people are more likely to connect to the individuals with similar rhythms). Perhaps, fixing a set of players with λ_i and letting them to form a network would tell us something about the optimal network in the presence of heterogeneity.

This is an excellent suggestion. Motivated by this comment, we have added a new paragraph (as below) to the Discussion Section of the revised manuscript.

“A natural extension of our findings is exploring the scenario with multiple strategies [56-58]. In this way, the diverse strategy update rhythms may couple multiple strategies with complex dynamics. In addition, our findings may contribute to the study of network formation, elucidating the factors influencing group formation, such as individuals' propensity to establish connections with those who share similar rhythms. Specifically, discovering the scenarios wherein individuals with similar update rates are allowed to construct a group may provide valuable information regarding the optimal network configuration in the context of heterogeneity.”

Minor comments:

1. I believe that the general statement in lines 31-33 is only true in the absence of other mechanisms (such as direct, indirect or network reciprocity, punishment, etc). Some references to a broader set of literature would be very much appreciated.

We fully agree with Reviewer #1. We have added *“without additional mechanisms including direct [17, 18] or indirect reciprocity [19-22], and punishment [23, 24]”* to lines 31-32 in the revised manuscript.

2. I think the section on “Theoretical analyses” jumps into technical details a bit abruptly. Perhaps, one has to motivate a bit more for why we would want to consider different distributions of λ and how can they be interpreted (what does it mean for the updating rule). Here, again, the question is how sensitive are the results wrt parameters used in Figure 5?

Motivated by this comment, we added *“In addition to the role of hubs that we uncover for the three specific update rate settings ($\lambda_i = 1/k_i$, $\lambda_i = 1$, $\lambda_i = k_i$), can we derive the general rule for promoting cooperation that also applies to other distributions of update rates?”* to present the motivation of considering different distributions of update rates at the beginning of this section (see lines 196-198).

To show the robustness of our results in Fig. 5, we further conducted more calculations for different parameters and presented the robustness of our results in Supplementary Fig. 4. We also added *“The robustness of our results with different average degrees is shown in Supplementary Fig. 4.”* to the caption of Fig. 5.

3. In the algorithm, what are the learning rate, decay rate and constant δ_{opt} ? how are they defined, why the values set this way and what is their effect? Perhaps, a more broader discussion of the algorithm in the main text would also be good.

We thank Reviewer #1 for this critical comment. To clarify the definition and effect of these parameters, we have added *“Note that the settings of the learning rate ϵ , decay rate ρ and constant δ_{opt} parameters are the same as those in RMSprop—the learning rate ϵ controls the step size of the iteration; ρ controls the decay rate of the moving average; and δ_{opt} is a small constant added to the denominator to prevent division by zero (see the values of those parameters in Methods)”* to lines 266-269 in the revised manuscript.

4. In the methods section, I think it would be beneficial to repeat previous notation descriptions (such as Λ , K or $F_j(\mathbf{x})$).

We thank Reviewer #1 for this excellent suggestion. We have revised the manuscript accordingly.

5. What is the reproductive-value-weighted frequency \hat{x} ?

We have added the definition of \hat{x} and added “where π_i is the reproductive value [62-64] uniquely solved by Supplementary equation (S2), quantifying the expected contribution of site i to the future gene pool under neutral drift.” in lines 363-364 of the revised manuscript.

Comments on the figures:

1. Figure 1: what are the update rates λ_i chosen for each player on the right?

Here $\lambda_1 = 1$, $\lambda_2 = 1.25$, $\lambda_3 = 1.75$, $\lambda_4 = 2$. We have updated Fig.1 accordingly.

2. Figure 2d: could you please add a row for $\lambda = k_i$?

This suggestion has been incorporated in the revised figure.

3. Figure 2e: I found it very hard to read the markers. Similar comment about Figure 5b.

We have updated Fig.2e and Fig.5b to enhance their readability.

4. Figure 5a: while there is certainly more variability in the results for ER, SM and SF networks, the average C^* are still kind of similar across different distributions. Can the authors comment on that?

Certainly. We calculated C^* for different distributions of update rates on the same network for each type, where each individual was randomly assigned a rate according to a given distribution repeatedly. When the number of random samples is large, the coupling of node degree and rate will not bring quantitatively different results. We clarified this point by adding “Note that the coupling of node degree and update rate will not bring quantitatively different results on the average C^* .” to the caption of Fig. 5.

Math notation comments:

1. I wonder if \mathcal{C}^* was used before or if the authors come up with this for the first time? Mathematically, $\mathcal{\{}}$ usually refers to sets, not to values of a variable. But if this is a common notation, let it be.

We have changed \mathcal{C}^* into C^* in the revised manuscript.

2. Why in the main text the edges are denoted by e_{ij} while in the algorithm the authors are using the adjacent matrix A and A_{ij} ? are these two different? Also, I personally believe that it is mathematically more elegant to denote vectors as small bold and matrices as capital non-bold letters.

Those two notations have the same meaning, and we have changed $A_{(i,j)}$ to $E_{(i,j)}$ (non-bold) in the revised Algorithm.

3. Am I correct in understanding that \mathbf{u} in equation (5) is used to denote a uniform draw? Why is it bold?

The reviewer is correct. To avoid confusion, we used the nonbold version in the revised manuscript.

In summary, we wish to thank Reviewer #1 for the valuable comments and suggestions. Addressing them has considerably improved the quality of our manuscript.

Response to Reviewer #2:

By calculating the fixation probability of a single cooperative mutant, the authors demonstrate that dynamical heterogeneity (in the form of differences in strategy update rates) can promote cooperation in structurally heterogeneous networks (e.g., scale-free, small world, or Erdos-Renyi). The authors find that the evolution of cooperation in heterogeneous networks depends crucially on the distribution of learning rates: if rates increase with a node's degree, this hinders the evolution of cooperation; if the reverse is true, cooperation evolves more easily. This result is used to develop an algorithm to find the optimal update rates to promote the emergence of cooperation on a given network.

This study provides new insights on the interplay between dynamical and structural heterogeneity and the evolution of cooperation and has the potential to make an important contribution to the literature on heterogeneity and cooperation. However, there are two major points that the authors need to address:

We thank Reviewer #2 for reviewing our manuscript and pointing out that our study provides new insights and has the potential to make an important contribution to the field. Next, we address each issue raised by the reviewer in order.

1) The first point concerns the lack of arguments about the evolution of optimal update rates. The authors demonstrate that certain distributions of update rates (i.e., when update rates decrease with increasing node degree) favour cooperation. But the question remains: why and under what conditions would natural selection promote such distributions of update rates? In other words, is it always advantageous for individual nodes to modulate their update rate so that it is inversely proportional to their degree?

If the answer to the above question is yes, then distributions of learning rate that promote cooperation will evolve. But it is also possible that individual nodes maximise their payoffs by increasing their learning rate proportionally to their number of connections: for example, if a cooperative hub has many cooperative neighbours, it could be advantageous (at least, in the short run) to change strategy to defection in order to exploit them.

If this is indeed the case, natural selection would not promote the evolution of distributions of learning rates that favour cooperation. The authors' results would be more relevant to understanding the behaviour of agent-based models than the evolution of biological populations. The relevance and robustness of the results would be enhanced by a clear explanation of why, and under what conditions, natural selection would lead to a distribution of learning rates that favours cooperation.

We thank Reviewer #2 for this very insightful comment. We fully agree with the reviewer that it is important to explain and clarify why it is advantageous for individuals to have update rates that are inversely proportional to degree. Here we would like to start with the meaningful example raised by the reviewer. In the example given, a cooperative hub with many cooperative neighbours could obtain a high payoff immediately by becoming a defector and exploiting its neighbours. However, we argue that this is detrimental to the *long-term* payoffs for the hub. The underlying mechanism is intuitively shown in Figs. 3 and 4 in the main text. When the update rate of each individual is proportional to its number of connections, the hub updates too fast to maintain the cooperative cluster (Fig. 3c in the main text) and leads to a low fraction of cooperative neighbours (Fig. 4g, h in the main text). Therefore, the frequent updates of a hub will bring itself a lower payoff over a relatively long period.

To qualitatively explain why, and under what conditions, natural selection would lead to a distribution of update rates that favours cooperation, we calculate the long-term payoffs for each individual in a general evolutionary process with a mutation rate u ($0 < u \leq 1$). The population starts with a single random cooperative mutant in a population with full defectors, following the evolutionary games in our main result, it will reach full cooperation or defection after a period of evolution. Then in full cooperation (defection), with probability u a single random defector (cooperator) is introduced; otherwise, with probability $1 - u$ the population stays in full cooperation

Figure R2: The advantage of infrequent strategy updates of hubs for accumulating individual long-term payoffs. We calculate the average long-term individual payoff $P_{\lambda_i=1/k_i}$, $P_{\lambda_i=1}$ and $P_{\lambda_i=k_i}$ corresponding to different settings of update rates ($\lambda_i = 1/k_i$, $\lambda_i = 1$, $\lambda_i = k_i$) during the evolutionary process of a population on a scale-free network. A single mutant arises uniformly with a mutation rate u when the population reaches full cooperation or full defection. **a**, We present the long-term average payoff of corresponding settings of update rates in a population with a high mutation rate ($u = 1$). The variance of the long-term average payoffs across different individuals is small, indicating that infrequent strategy updates of hubs are beneficial to all individuals. Furthermore, we show the payoff difference $P_{\lambda_i=1/k_i} - P_{\lambda_i=1}$ between the settings with update rates inversely proportional to degrees ($\lambda_i = 1/k_i$) and identical rates ($\lambda_i = 1$) in **b**. Analogously, the long-term payoff difference between identical rates and rates proportionally to nodes' degrees $P_{\lambda_i=1} - P_{\lambda_i=k_i}$ is shown in **c**. We present that $P_{\lambda_i=1/k_i} > P_{\lambda_i=1}$ and $P_{\lambda_i=1} > P_{\lambda_i=k_i}$ over different mutation rates and selection intensities. The long-term payoff in (a-c) are obtained over 10^3 independent runs, where the average payoff in each run is obtained over 10^6 rounds.

(defection) [50]. Specifically, when the mutation is rare ($u \rightarrow 0$), the population is almost always in full cooperation or full defection, and the time spent in full cooperation (defection) state is proportional to ρ_C (ρ_D) [50]. In this scenario, the distribution of the update rates which promotes cooperation more will lead to a higher long-term average payoff, since individuals have a payoff $b - c$ in full cooperation but 0 in full defection. Even if the mutation rate is high ($u = 1$), the settings of $\lambda_i = 1/k_i$ still leads to higher long-term payoffs for all individuals (Fig. R2a).

Our results further indicate the advantage of infrequent updates of hubs holds over a wide range of selection intensities and mutation rates (Fig. R2b, c). For different selection intensities and mutation rates, we found that the inverse relationship between update rate and node degree ($\lambda_i = 1/k_i$) always leads to a higher long-term payoff than $\lambda_i = 1$ and $\lambda_i = k_i$. And the payoff difference increases when the mutation rate decreases. Therefore, natural selection would promote the evolution of update rates which are inversely proportional to the node's degree when individuals maximise their long-term payoff during the evolutionary process.

Motivated by Reviewer #2's important comment, we added Fig. R2 to Fig. 4 in our revised manuscript and clarified this point by adding a new paragraph (as below) at the end of the Section "Role of network hubs".

"Furthermore, we find that infrequent updates of hubs can also bring long-term advantages to individuals. We could even consider the general evolutionary process with mutation, where a mutant appears with probability u when the population reaches full cooperation or full defection. Each individual accumulates long-term payoffs during a long period of time. Even with a high mutation rate ($u = 1$), we show that the inverse relationship between update rates and nodes' degrees results in a higher long-term payoff for individuals than identical rates ($\lambda_i = 1$) (Fig. 4i). In contrast, frequent updates of hubs ($\lambda_i = k_i$) lead to a lower payoff than the identical settings (Fig. 4i). This result is also robust over different mutation rates and selection intensities (Fig. 4j, k). Moreover, when the mutation is rare, the population is almost always in full cooperation or full defection, and the time spent in full cooperation (defection) is proportional to ρ_C (ρ_D) [50]. Therefore, the settings of update rates which promote cooperation further lead to a higher long-term average payoff, since individuals get $b - c$ in full cooperation but 0 in full defection."

2) The second point concerns the significance of the work in relation to the existing literature. I disagree with the authors' claim that their result (that heterogeneous update rates promote cooperation) is “surprising” and “undercuts the conventional wisdom that heterogeneous structure is generally antagonistic to cooperation”. The main study cited to support this claim is Allen et al. (2017). Allen et al.'s find an exact formula that expresses the critical benefit to cost ratio $(b/c)^*$ above which cooperation will evolve as a function of coalescence time. The degree distribution does not figure explicitly in it, and it is not very obvious to me that a more heterogeneous network will automatically lead to a higher $(b/c)^*$.

This is an important point that does warrant clarification. In response, we recapped the results of Fig.4a in Allen et al. [5] and plotted the critical benefit-to-cost ratio of scale-free and random regular networks in Fig. R3a and add it to Supplementary Fig. 13. We confirm that highly heterogeneous scale-free networks require a higher threshold for favouring cooperation than the homogeneous random regular networks over different mean degrees.

Figure R3: Effect of different update rules on the evolution of cooperation. **a**, Scatter plot of the critical benefit-to-cost ratio C^* versus mean degree $\langle k \rangle$ of scale-free (orange dot) and random regular (blue dot) networks with $N = 100$. **b**, Regarding the frequency of cooperators in the population starting from equal cooperators and defectors [6], scale-free networks (orange) lead to a lower frequency of cooperation compared to random regular networks (blue) under death-birth update. **c**, The cooperators x (blue) with all defective neighbours (red) obtains $P_x = 0$, and the defectors y with k_y cooperative neighbours obtains $P_y = k_y b$. Under the update rule in the series of studies by Santos et al. [6], whenever x is updated and y is randomly chosen to be imitated, individual x imitates the strategy of y with probability k_y/k_x .

However, the landmark paper by Allen et al. is not the only precedent that has analysed the role of degree (distribution) in evolutionary game theory. Indeed, in Fotouhi et al. [31], the critical ratio C^* is analytically approximated by

$$C^* \approx \frac{N-2\langle k^2 \rangle / \langle k \rangle^2}{N/\langle k \rangle - 2\langle k^2 \rangle / \langle k \rangle^2},$$

where $\langle k \rangle$ is the mean degree and $\langle k^2 \rangle$ is the second moment of the degree distribution (equation (4.5) in [31]). The presence of the second moment $\langle k^2 \rangle$ here codifies a role of degree distribution shape beyond average degree. In particular, the higher $\langle k^2 \rangle$, the more heterogeneous the degree distribution. Indeed, for homogeneous networks wherein all nodes have the same degree, this expression reduces to

$$C^* \approx \frac{N-2}{N/\langle k \rangle - 2}$$

since $\langle k^2 \rangle = \langle k \rangle^2$. In contrast, for heterogeneous networks, we will have $\langle k^2 \rangle > \langle k \rangle^2$ since the variance of the degree distribution is always greater than 0, which will in turn lead to a larger critical ratio C^* than in homogeneous networks.

On the contrary, previous studies directly comparing homogeneous and scale-free networks revealed that a heterogeneous degree distribution promotes cooperation, both in the prisoner's dilemma (Santos and Pacheco, 2005; Santos, Rodrigues, and Pacheco, 2006) and in other social dilemmas (stag hunt and snowdrift; Santos, Pacheco, and Lenaerts, 2006). There are also other studies indicating that heterogeneity (in the form of random payoff fluctuations) is a powerful force in maintaining cooperation in structured populations (see, for example, Javarone and Amaral, 2020).

While the studies above do not explicitly calculate fixation probabilities, but only the average frequency of cooperators, it seems important to clarify why these two approaches lead to different conclusions. The authors acknowledge this difference in the discussion, but to me they do not satisfactorily explain why the difference exists in the first place.

Regarding the above literature quantifying the evolution of cooperation by the average frequency of cooperators, Reviewer #2 is absolutely correct. We will clarify why the difference exists in the first place together in the response after the next paragraph.

One possible explanation might be that, in the update function used in the Santos and Pacheco (2005) and the two Santos et al. (2006) studies, the probability that node x imitates the strategy of node y is proportional to the ratio between payoff differences and the maximum degree of either node. Therefore, it is easier for a high-payoff strategy to spread between nodes with a low degree than to or from a hub. Is this analogous to one of the conditions studied in this manuscript (lower update rates for hubs)? The authors will certainly have a better sense of whether these two mechanisms are similar or not; perhaps this should be addressed in the discussion.

We thank Reviewer #2 for this insightful comment. The simple answer is that, the two approaches have different update rules. Indeed, if we take the death-birth update rule that we used in our manuscript to calculate the average frequency of cooperators starting from an equal number of cooperators and defectors, we still conclude that heterogeneous networks impede the emergence of cooperation than the homogeneous counterpart (Fig. R3b). That is, the update rule determines the conclusions about which structure is more advantageous in promoting the emergence of cooperation.

From the perspective of the microscopic mechanism, we think that the reason why heterogeneous degree distributions promote cooperation in the studies of Santos *et al.* [6, 33, 34] is because hubs tend to have infrequent strategy switching (as correctly observed by the reviewer). In this series of studies, an individual x imitates the strategy of node y with probability $(P_y - P_x)/(k_{>}(T - S))$ if node y 's payoff P_y is higher than that of node x (P_x), and $k_{>}$ indicates the maximum degree of either node. There are two main reasons that this leads to infrequent strategy switching of hubs. The first is that these studies use accumulated payoff obtained from all neighbours, which naturally leads to the high payoff of hubs, thus lowering the probability that a hub will have any neighbour with higher payoff to imitate. The second reason is that the payoff difference is normalised by $k_{>}(T - S)$, which is much larger than the payoff difference that a smaller node can have higher than a hub.

In Fig. R3c, we provide a simple example to explain how this update rule is advantageous for the emergence of cooperation in heterogeneous networks. Here we follow the parameter setting of Santos *et al.* ($T = b, R = 1, P = S = 0$). The hub node x with k_x neighbours obtains payoff $P_x = 0$ because all its neighbours are defectors, and defective neighbour y accumulates payoff $P_y = k_y b$. Whenever x is updated and y is randomly chosen to be imitated, the probability that x copies the strategy of y is k_y/k_x . Therefore, a cooperative hub still can survive even if all its neighbours are defectors. In contrast, under death-birth update, the probability for x to change to defection is 1. This implies that the update rule applied by Santos *et al.* [6] naturally leads to infrequent updates of hubs, making it easier for a high-payoff strategy to spread between low-degree nodes than to a hub. And this is consistent with the underlying mechanism studied in our manuscript, namely when we consider the evolutionary success of cooperators, a low update rate for hubs promotes the evolution of cooperation.

Motivated by Reviewer #2's observations, we now compare these two mechanisms in a new paragraph (see below) in Discussion, supplemented by a more comprehensive discussion in the new Supplementary Section 5. We also point out both the results by Allen *et al.* and the following study by Fotouhi *et al.* [31] have informed that heterogeneous structure is generally antagonistic to cooperation in the revised line 44.

"From the perspective of microscopic mechanism, we unveil that different update rules render the conflict results. Regarding the frequency of cooperators, previous canonical framework and update rule naturally lead to

infrequent strategy switching (Supplementary Section 5) [6,33,34]. This facilitates the formation of cooperative clusters and leads to a high fraction of cooperators on heterogeneous networks. Previous findings are consistent with the underlying microscopic mechanism in our study, namely infrequent updates of hubs facilitate the emergence of cooperation. Indeed, by applying the canonical death-birth update with identical rates in the framework analysing the frequency of cooperators [6, 33, 34], we find that heterogeneous networks impede the average frequency of cooperators compared to homogeneous scenarios (Supplementary Fig. 13)."

To summarise: it seems to me that the novelty of this study lies not in the surprising nature of the main results, but rather in its potential to provide a unifying framework to understand the interplay between structural heterogeneity, learning rates, and cooperation. To better integrate their work into the existing literature, the authors should address in more detail why different approaches have yielded different conclusions, acknowledging the existing literature that links heterogeneity with cooperation, and perhaps comparing their results with experimental studies on cooperations in heterogeneous networks (e.g., Gracia-Lázaro et al., 2012).

We thank Reviewer #2 for saying that our work provides a unifying framework to understand the interplay between structural heterogeneity, update rates, and cooperation. We stressed this point in the revised abstract by adding *"Our findings provide a unifying framework to understand the interplay between structural heterogeneity, behavioural rhythms, and cooperation."* And we further clarified why different approaches have yielded different conclusions in the Discussion section. We also thank Reviewer #2 for bringing the important experimental study to our attention. Indeed, by behavioural experiments, Gracia-Lázaro *et al.* reported that heterogeneous networks do not promote the emergence of cooperation. The reason is that the actions of players are relevant to the level of cooperation in their neighbourhood and the actions in the previous round rather than their observed payoffs of neighbours.

Motivated by this meaningful comment, we have added a new paragraph (as below) in our revised Discussion section.

"Furthermore, we compare our results with experimental studies on cooperation in heterogeneous networks. Consistent with our theoretical findings, there is an insightful experimental study also reporting that heterogeneous networks do not promote cooperation in Prisoner's Dilemmas [55]. In this behavioural experiment, a player's decisions to cooperate or defect are relevant to the level of cooperation in their neighbourhoods, which renders the network irrelevant. Therefore, the main difference between this experimental finding and our study lies in the update rules. Specifically, players are more likely to imitate the strategy from neighbours with higher payoffs in our theoretical framework. To further uncover the behavioural dynamics from the perspective of fixation probability, a promising future application involves the design of human behavioural experiments starting from a single cooperator and ending with full cooperation or defection. Comparing the individual behavioural mode in experiments from these two perspectives will facilitate the understanding of the emergence of cooperation in realistic scenarios."

Finally, the manuscript would benefit from a few minor clarifications:

3) I find the word "arbitrary" potentially confusing in the context of learning rates. It might give the impression that each node could *arbitrarily* update its learning rate (e.g., through some evolutionary dynamics), which is obviously not the case. The learning rates are different from node to node, but fixed for the duration of a simulation.

We have changed the word "arbitrary" to "diverse" or "personalised" in the revised manuscript to avoid ambiguity. Moreover, motivated by the constructive comment, we further changed the paper's title accordingly.

4) The authors might want to clarify that they are using a simultaneous donation game to model the evolution of cooperation and that this is a specific type of prisoner's dilemma; the two are not equivalent. I'd be interested if the authors have a sense of whether their general conclusions would also apply to more general forms of the prisoner's dilemma and, more broadly, to other social dilemmas as well.

We thank Reviewer #2 for this comment. We have now studied general social dilemmas by deriving the general formula of critical benefit-to-cost ratio to cover other types of games (including snowdrift and stag hunt games) beyond the donation game (Supplementary Section 3.5). Furthermore, we bridged the result under other games to the condition for favouring cooperation under the donation game, and found that our general conclusion is robust to the choice of game.

In a general two-player game, a cooperator receives “rewards” R from mutual cooperation, while defectors obtain “punishment” P from mutual defection. A defector attempting to exploit a cooperator obtains T and leaves S to its opponent cooperator. We found that cooperation is favoured over defection when $R > P + (T - S)(C^* - 1)/(C^* + 1)$, where a lower threshold for R can be achieved with a lower C^* . This indicates that our conclusion under the donation game can be completely applied to different social dilemmas, such as the general prisoner’s dilemma ($T > R > P > S$), snowdrift game ($T > R > S > P$) and stag hunt game ($R > T \geq P > S$). Therefore, the inverse relationship between update rates and node’s degree can facilitate the emergence of cooperation in different social dilemmas.

Motivated by Reviewer #2’s comment, we added this point as Supplementary Section 3.5 in the revised SI. Moreover, we added a new paragraph (as below) at the end of the section “A simple condition for the emergence of cooperation” in the revised manuscript.

“Moreover, we show that this general conclusion can also be applied to other social dilemmas (Supplementary Section 3.5). For the general two-player game, a cooperator receives “rewards” R from mutual cooperation, while defectors obtain “punishment” P from mutual defection. A defector attempting to exploit a cooperator obtains T and leaves S to its opponent cooperator. We show that cooperation is favoured over defection when $R > P + (T - S)(C^ - 1)/(C^* + 1)$, where a lower threshold for R can be achieved with a lower C^* . Note that here C^* is exactly the critical threshold under the donation game. This indicates our conclusion applies to other social dilemmas, such as the general prisoner’s dilemma ($T > R > P > S$) [16], snowdrift game ($T > R > S > P$) [3] and stag hunt game ($R > T \geq P > S$) [53].”*

5) Abstract: the sentence “Combining theoretical and computational techniques with synthetic and empirical data analyses” is potentially misleading, as the paper does not present or analyse any empirical data. In fact, I think it would be very interesting to compare the results of this study with empirical findings on cooperation in heterogeneous networks.

We thank Reviewer #2 for alerting us to this potential confusion. We have changed the sentence to “*Combining analytical and numerical calculations on synthetic and empirical networks*” in the revised Abstract. Moreover, motivated by the reviewer’s comment about comparing our results with empirical findings, we have added this point as an additional paragraph in the revised Discussion section.

REFERENCES

Santos, F. C., & Pacheco, J. M. (2005). Scale-free networks provide a unifying framework for the emergence of cooperation. *Physical review letters*, 95(9), 098104.

Santos, F. C., Rodrigues, J. F., & Pacheco, J. M. (2006). Graph topology plays a determinant role in the evolution of cooperation. *Proceedings of the Royal Society B: Biological Sciences*, 273(1582), 51-55.

Santos, F. C., Pacheco, J. M., & Lenaerts, T. (2006). Evolutionary dynamics of social dilemmas in structured heterogeneous populations. *Proceedings of the National Academy of Sciences*, 103(9), 3490-3494.

Amaral, M. A., & Javarone, M. A. (2020). Heterogeneity in evolutionary games: an analysis of the risk perception. *Proceedings of the Royal Society A*, 476(2237), 20200116.

Gracia-Lázaro, C., Ferrer, A., Ruiz, G., Tarancón, A., Cuesta, J. A., Sánchez, A., & Moreno, Y. (2012). Heterogeneous networks do not promote cooperation when humans play a Prisoner's Dilemma. *Proceedings of the National Academy of Sciences*, 109(32), 12922-12926.

We thank Reviewer #2 for pointing out those references, which have been fully cited in the revised manuscript.

In summary, we wish to thank Reviewer #2 for his/her thorough and constructive review of our manuscript. The comments therein have considerably improved the paper.

Response to Reviewer #3:

The paper focuses on the evolution of cooperation in networks. Typically, in network models, the update rate of nodes is fixed or depends on a stochastic process, which is uniform across all nodes. This paper introduces the concept of heterogeneous strategy update rates as a factor that could facilitate the evolution of cooperation. It provides a theoretical framework and derives its results both analytically and through simulations.

The paper is well-written, engaging, and to the point. The clarifications and examples are very helpful for readers. I enjoyed reading the paper.

We thank Reviewer #3 for reviewing our manuscript and the overall positive assessment. Next, we address each issue raised by the reviewer in order.

My main concern regards the relevance and generality of its main results. The paper demonstrates the effect of non-uniform update rates in a very specific setting: when they are inversely proportional to the average connectivity of a node. My suggestion is to address several important questions: Why should we expect update rates to differ, and to what extent? Are randomly distributed update rates (e.g., following a Poisson process) incorrect or insufficient for modeling and approximating the nature of different updating events? If so, why should we expect a negative correlation between connectivity and update rate? I believe the introduction overlooks these crucial questions. The paper could be strengthened by adding real-world relevance to this modification. For example, one might argue that a higher number of connections could slow down the updating process due to more information needing processing. However, the results are intriguing in the opposite case: when update rates vary inversely with the number of connections. This not only seems like a very specific case but also appears to be an unlikely scenario (correct me if I am wrong and my apologies if that's the case). The authors also provide real-world network examples (office, student, Attiro family contacts, and San Juan family contacts) and demonstrate how the critical benefit-to-cost ratio would change based on the assumption of update rates. Justifying the main result for these networks would be beneficial. Without solid real-world relevance, the paper remains interesting but might be better suited for a specialist journal.

We thank Reviewer #3 for this critical comment. We would like to take the chance to clarify the real-world relevance of our work. An empirical study on evolutionary games has found that individuals adopt many different possibilities for strategy updating in human behavioural experiments [42]. As keenly noted by the reviewer, a node with a high number of connections may need more time to process the information. Indeed, both cognitive processing speed and personality traits can impact individuals' decision-making time.

In the real world, humans make decisions in more sophisticated ways than is captured by simple rules like synchronous updating in many theoretical studies and computer simulations. From the perspective of information processing speed, previous empirical studies have indeed found that individuals vary significantly in cognitive processing speed [43-45]. For example, individuals with greater cognitive abilities display a greater information processing speed, measured by a short reaction time. Beyond that, personality traits also affect decision-making speed [46]. Based on the empirical evidence on individual differences in information processing and decision-making speed, we argue that the previous assumption of identical update rates for all individuals is too ideal to model the updating event in realistic scenarios. This recognition is in fact a key motivation of our study.

In this work, we systematically studied the impact of diverse update rates on the evolution of cooperation, including different rate distributions and both negative and positive proportional relationships with individual connections. We concluded that the emergence of cooperation is promoted when the update rate is inversely proportional to degree. We also analysed real network data which may contain some specific interaction patterns [47, 48] to get closer to real interactions. Moreover, we found that our conclusion can offer insights into realistic scenarios of decision-making. For example, if popular individuals (i.e., hubs with many social connections) in a social network frequently change their opinions, it is not conducive to reaching a consensus in the group, so they may need to update their opinions less frequently to drive the emergence of collective behaviours.

Motivated by the reviewer's comment, we emphasized the real-world relevance as a motivation of our study by adding a new paragraph (as below) to the Introduction of the revised manuscript.

“In reality, humans behave in more sophisticated ways in decision-making than simple identical updating. An empirical study on evolutionary games uncovered that individuals are observed to have many different possibilities for strategy updating in human behavioural experiments [42]. Indeed, both cognitive processing speed and personality traits can have an impact on the time of individual decision-making. Previous empirical studies have found that individuals vary significantly in cognitive processing speed [43-45]. For example, individuals with greater cognitive abilities have high information processing speed and display a short reaction time. On the other hand, many personality traits are also evidenced to correlate with the decision-making speed [46]. Taken together, the previous assumption of identical update rates for all individuals is too ideal to portray the updating event and heterogeneous individual interaction rhythms in realistic scenarios [47, 48]. This prompts us to ask how this dynamical heterogeneity might interact with structural heterogeneity to alter the evolution of cooperation.”

Related to this, the abstract suggests that this update rate extension can be relevant for design purposes. The authors went out of their way to provide an algorithm to optimize collective cooperation by changing individuals' update rates. (Thanks also for the Matlab code.) Although that's very interesting, the same question of behavioral relevance arises. Update rates seem to me a highly endogenous aspect. It's challenging to think of many ways to intervene in the update rates in real-world scenarios (as opposed to changing the network structure, for instance). Elaborating on this would be immensely helpful.

We agree that more discussion on the realism of optimization/global interventions is warranted. As the reviewer says, it is indeed challenging to intervene to change human behaviour in real-world scenarios, and the focus of our study is not on exerting control over individuals. However, centralized design of update rates *is* feasible in certain built systems. For example, our algorithm can be applied to designing autonomous systems (e.g. behaviour-based formation of swarm robots) and finding optimal update rates from a global perspective.

We have clarified this point by adding *“As an engineering application of designing unmanned and autonomous systems, can we adopt the simple heuristic to favour collective cooperation among agents? Specifically, can we find the optimal set of λ_i for a given networked system?”* (see main text lines 260-262) at the beginning of Section *“The optimal update rate on any network”*.

Another aspect is the paper's limited connection to previous studies dealing with update mechanisms and rates. I am not an expert of this particular topic so it might be me. But it's somewhat difficult to ascertain the state of the literature in this area. It would help, especially for interdisciplinary readers, if the paper could situate itself within the context of other studies on heterogeneous update rates, if any exist.

I leave the judgment and decision to the authors, but here are some studies that caught my attention:

Allen, James M., and Rebecca B. Hoyle. 'Asynchronous Updates Can Promote the Evolution of Cooperation on Multiplex Networks'. *Physica A: Statistical Mechanics and Its Applications* 471 (April 2017): 607–19.

Grilo, Carlos, and Luís Correia. 'Effects of Asynchronism on Evolutionary Games'. *Journal of Theoretical Biology* 269, no. 1 (January 2011): 109–22.

———. 'The Influence of the Update Dynamics on the Evolution of Cooperation'. *International Journal of Computational Intelligence Systems* 2, no. 2 (June 2009): 104–14.

Johnson, Tim, and Oleg Smirnov. 'Temporal Assortment of Cooperators in the Spatial Prisoner's Dilemma'. *Communications Biology* 4, no. 1 (12 November 2021): 1283.

Wang, Dongqi, Xuanyue Shuai, Qihui Pan, Jingye Li, Xiaolong Lan, and Mingfeng He. ‘Long Deliberation Times Promote Cooperation in the Prisoner’s Dilemma Game’. *Physica A: Statistical Mechanics and Its Applications* 537 (January 2020): 122719.

Zhang, Jianlei, Chunyan Zhang, Ming Cao, and Franz J. Weissing. ‘Crucial Role of Strategy Updating for Coexistence of Strategies in Interaction Networks’. *Physical Review E* 91, no. 4 (2 April 2015): 042101.

We thank Reviewer #3 for providing this list of important references. Of course, it is imperative that we situate our findings within the context of other studies on update mechanisms and rates, especially the synchronous and asynchronous updating in the Introduction. We have rewritten the third paragraph of the Introduction as follows.

“Despite remarkable advances in our understanding of the emergence of cooperation, many studies have confined that individuals update their strategy synchronously [6, 33-35]—that all individuals update at exactly the same time. However, the perfect synchronism is absent from the real world, and it has been shown that asynchronous updating—individuals are allowed to update at different time—can significantly alter the evolution of cooperation [36-40]. A typical asynchronous update rule is the death-birth update, where only a single individual is selected uniformly at random to die and their neighbours spread their strategies by competing for the vacant position at each time step [4, 5]. Alternatively, individuals may change their strategies by mimicking that of their neighbours (imitation [4], pairwise comparison [12, 41]). All these important canonical updating rules have been based on a key assumption: that all individuals update their strategies at the same rate.”

Minor Comments:

- The abstract mentions (line 20): “Combining theoretical and computational techniques with synthetic and empirical data analyses, we find that when individuals’ update rates vary inversely with their number of connections, heterogeneous connections actually outperform homogeneous ones in promoting cooperation.” If I’m not mistaken, the empirical data analyses do not support or refute the claim but rather demonstrate the application of the logic. So, it might be clearer to specify this, whether I am right or wrong in my understanding.

In the revised abstract, we have changed the sentence in question to “*Combining analytical and numerical calculations on synthetic and empirical networks*” in order to avoid potential confusion.

- I really appreciate the effort the paper puts into providing intuition. Regarding the “lock-in” explanation of the mechanism (line 122), if you could also explain why this effect is not symmetric for cooperation and defection, and whether it favors cooperation, that would assist the reader.

We thank Reviewer #3 for this excellent suggestion. We would like to take this chance to clarify more in detail regarding why defectors cannot benefit from this “lock-in” effect. The essential reason is that cooperators can obtain higher payoffs (mutual cooperation brings $b - c$ to both players) through cooperative clusters to resist the invasion of defectors, which facilitates the fixation of cooperation in the population. In contrast, the “lock-in” effect for a defector on hubs can indeed drive its neighbours to defectors, but it will leave zero payoff for defectors (mutual defection leaves nothing to both players). In this way, cooperators can benefit from the “lock-in” effect by forming stable cooperative clusters, yet defectors further reduce survival chances due to the low payoffs.

Motivated by this comment, we have added the sentence “*Note that this “lock-in” effect can facilitate the formation of cooperative clusters to have higher payoffs to resist the invasion of defectors, yet defectors receive a lower payoff after driving their neighbours to defectors and further reduce their survival chances.*” to lines 140-143 of the revised manuscript.

Happy to read the paper and I hope my comments make sense to the authors.

We wish to thank Reviewer #3 for his/her tremendous efforts in reviewing our manuscript. These comments have markedly improved the clarity and relevance of our work to a broad audience.

Reviewers' Comments:

Reviewer #1:

Remarks to the Author:

The authors have addressed my comments.

Reviewer #2:

Remarks to the Author:

I am satisfied with how the authors have addressed my previous comments. I believe that the manuscript is now suitable for publication in Nature Communications, and am grateful for the opportunity of reviewing such an interesting and insightful study.

Reviewer #3:

Remarks to the Author:

I want to thank the authors for addressing all my comments and revising their manuscript thoroughly.

As mentioned earlier, the manuscript is interesting as it shows a novel approach to cooperation on networks. I believe it will appeal to a wider multidisciplinary audience.

But finally, I must point out that the replication package still falls slightly below the standards. I understand the authors were likely focused on the manuscript, but an update to the repository would be appropriate. I'll list my comments in the relevant section of the reviewer assessment.

Response to Reviewer #1:

The authors have addressed my comments.

We thank Reviewer #1 for reviewing our manuscript again.

Remarks on code availability:

I believe the codes could benefit from comments on each line or code segment.

We thank Reviewer #1 for this constructive comment. Motivated by this comment, we have added comments for each line or code segment in the updated version of our codes.

In summary, we wish to thank Reviewer #1 for the valuable suggestions. Addressing them has considerably improved the quality of our codes.

Response to Reviewer #2:

I am satisfied with how the authors have addressed my previous comments. I believe that the manuscript is now suitable for publication in Nature Communications, and am grateful for the opportunity of reviewing such an interesting and insightful study.

We thank Reviewer #2 for reviewing our manuscript again and the previous insightful comments and suggestions.

Response to Reviewer #3:

I want to thank the authors for addressing all my comments and revising their manuscript thoroughly.

As mentioned earlier, the manuscript is interesting as it shows a novel approach to cooperation on networks. I believe it will appeal to a wider multidisciplinary audience.

We thank Reviewer #3 for reviewing our manuscript again and the continuous positive assessment.

But finally, I must point out that the replication package still falls slightly below the standards. I understand the authors were likely focused on the manuscript, but an update to the repository would be appropriate. I'll list my comments in the relevant section of the reviewer assessment.

Remarks on code availability:

Thank you once again for preparing the repository. Let me provide feedback based on the checklist:

- "A small (simulated or real) dataset to demo the software/code."

I am also uncertain if this is strictly necessary for simulation studies if they are brief. However, if the simulations take longer than a couple of minutes, providing a small output file would be appropriate for someone looking to replicate your findings without investing too much time.

Good point. We have provided the output file for the simulation code in the updated repository.

- "All software dependencies and operating systems."

For the Python file, a requirements.txt is necessary. I had to install a few packages before I could run the code. Even then, I encountered issues because the Python code utilized deprecated functions of the numpy and networkx packages. Having software dependencies listed in the requirements file would probably solve those issues.

We thank Reviewer #3 for this excellent suggestion. We have provided a requirements.txt which contains the list of package dependencies in the updated repository.

"Instructions" & "Expected Output."

Overall, more explicit instructions could enhance the usability of the repository. With the current format, I am uncertain about the type of output to expect from the simulations and how to interpret it. Clearer instructions would be beneficial, providing guidance on reproducing your main plots or any specific steps to follow.

We thank Reviewer #3 for this constructive comment. We have added detailed code comments and rewritten the README.md to provide clear instructions and guidance on reproducing our main plots. Inputs and Outputs for all programs are now clearly defined and interpreted in the updated README.md.

We wish to thank Reviewer #3 for his/her tremendous efforts in reviewing our manuscript and codes again. These comments have markedly enhanced the usability of our codes.